# One System, Two Rules: Asymmetrical Coupling of Speech Production and Reading Comprehension in the Trilingual Brain

**DOI:** 10.3390/brainsci15121288

**Published:** 2025-11-29

**Authors:** Yuanbo Wang, Yingfang Meng, Qiuyue Yang, Ruiming Wang

**Affiliations:** 1School of Psychology, Fujian Normal University, Fuzhou 350117, China; 2018010223@m.scnu.edu.cn; 2Philosophy and Social Science Laboratory of Reading and Development in Children and Adolescents, Ministry of Education, and Center for Studies of Psychological Application, School of Psychology, South China Normal University, Guangzhou 510631, China; 3School of Educational Science and Technology, Anshan Normal University, Anshan 114007, China

**Keywords:** trilingualism, age of acquisition (AoA), social usage, cross-modal priming

## Abstract

**Background/Objectives:** The functional architecture connecting speech production and reading comprehension remains unclear in multilinguals. This study investigated the cross-modal interaction between these systems in trilinguals to resolve the debate between Age of Acquisition (AoA) and usage frequency. **Methods:** We recruited 144 Uyghur (L1)–Chinese (L2)–English (L3) trilinguals, a population uniquely dissociating acquisition order from social dominance. Participants completed a production-to-comprehension priming paradigm, naming pictures in one language before performing a lexical decision task on translated words. Data were analyzed using linear mixed-effects models. **Results:** Significant cross-language priming confirmed an integrated lexicon, yet a fundamental asymmetry emerged. The top-down influence of production was governed by AoA; earlier-acquired languages (specifically L1) generated more effective priming signals than L2. Conversely, the bottom-up efficiency of recognition was driven by social usage frequency; the socially dominant L2 was the most receptive target, surpassing the heritage L1. **Conclusions:** The trilingual lexicon operates via “Two Rules”: a history-driven production system (AoA) and an environment-driven recognition system (Social Usage). This asymmetrical baseline challenges simple bilingual extensions and clarifies the dynamics of multilingual language control.

## 1. Introduction

A central challenge in psycholinguistics is elucidating the cognitive architecture that enables multilingual individuals to manage multiple linguistic systems simultaneously. It is now widely accepted that lexical access in multilinguals is language non-selective, meaning that words from all known languages are co-activated even when the context requires only one [1,2,3]. However, this co-activation is rarely symmetrical. A pervasive finding in bilingual research is an asymmetry in cross-linguistic influence, where the first language (L1) typically exerts a stronger priming effect on the second language (L2) than vice versa [4,5]. This phenomenon has generated a fundamental theoretical debate regarding the mechanisms driving this architecture: is the multilingual lexicon structured by the history of acquisition, or is it dynamically modulated by current usage?

Understanding these asymmetrical activation patterns is crucial for mapping the “control mechanisms” of the multilingual cognitive architecture [6,7,8]. While bilingual studies have provided the foundation for these questions, they often face a methodological bottleneck. In bilinguals, the “dominant” language is usually both the native language (L1) and the most frequently used one. This confound makes it difficult to disentangle the effects of Age of Acquisition (AoA) from Social Usage Frequency. To resolve this dilemma, we must look beyond the bilingual model. The present study leverages a trilingual population to dissociate these factors. Specifically, we aim to determine whether the flow of activation from speech production to reading comprehension is governed by the static stability of early acquisition [4] or the dynamic accessibility of current social use [9,10].

To build this trilingual map, we must first turn to the foundational principles and debates established in the more extensively studied bilingual domain. A widely accepted principle is that lexical access is language non-selective, meaning words from all known languages are co-activated, even in single-language contexts [1,2]. A critical feature of this system is its asymmetry, where cross-language priming is typically stronger from the first language (L1) to the second (L2) than in the reverse direction. This phenomenon has prompted a central theoretical debate regarding its primary driver. On one side, developmental models, such as the Revised Hierarchical Model (RHM), attribute this asymmetry to the trajectory of language acquisition [4,5]. On the other side, usage-based models like Multilink argue that current language dominance and frequency of use are more potent modulators [9]. This view is supported by findings that L2 use is a better predictor of L2→L1 priming than proficiency alone [10].

To resolve this theoretical tension, empirical work must explicitly disentangle the often-confounded effects of static proficiency and dynamic daily exposure. A compelling test was conducted by Chaouch-Orozco and colleagues in a study of late Spanish–English bilinguals, specifically designed to achieve this separation [11]. Methodologically, the study leveraged a participant sample where proficiency and exposure were only weakly correlated. Using continuous measures for both L2 proficiency (Oxford Quick Placement Test) [12] and daily L2 use (Dominance Scale; [13]), the researchers employed linear mixed-effects models to pinpoint the critical variable.

This careful methodological separation proved decisive. The results showed that a statistical model testing the influence of L2 proficiency on priming was non-significant. In stark contrast, a model incorporating L2 exposure revealed that a significant L2-to-L1 priming effect was present only for participants who reported high levels of daily L2 use. This specific pattern provides a clear verdict, challenging the classical RHM assumption and lending strong support to the usage-based Multilink model. As Chaouch-Orozco et al. concluded, “L2 exposure/use was the only experiential variable to show considerable influence on L2→L1 priming,” establishing it as the key modulator of cross-linguistic activation [11].

This conclusion, while clarifying the bilingual dynamic, simultaneously complicates the picture for trilingualism. The introduction of a third language (L3) creates a vastly more complex network of potential interactions, raising a central theoretical debate: is trilingual processing merely a quantitative extension of bilingual mechanisms, or does it involve qualitatively unique characteristics [14,15]? This question is impossible to address within a bilingual paradigm. In contrast, trilingualism allows us to investigate critical new questions, such as whether the influence of an L3 on an L2 operates differently than that of an L1. The complexity of L3 acquisition is evident in the diverse models proposed to explain its trajectory. These models implicate a wide range of factors, from the persistent influence of the L1 to the privileged role of the L2 and the variable of typological proximity (e.g., refs. [16,17]). Ironically, the methodological rigor of bilingual studies, which often exclude speakers of additional languages, has created a significant blind spot, leaving the ubiquitous global phenomenon of trilingualism profoundly under-researched [18].

To address this gap, the present study leverages a unique population of Uyghur (L1)–Chinese (L2)–English (L3) heritage speakers, which allows for the crucial dissociation of acquisition order from language dominance. This group offers several distinct advantages. First, their three languages are typologically distinct (Turkic, Sino-Tibetan, and Indo-European), helping to isolate cognitive effects from structural overlap. Second, their sociolinguistic context is highly informative. Specifically, Chinese (L2) serves as the dominant societal language and the medium of instruction for English (L3). This unique configuration creates a natural laboratory to test hypotheses like the “L2 Status Factor” [16]. Foundational work suggests that when L2 is the medium of instruction for L3, it forges strong L2–L3 lexical pathways. This unique context allows us to investigate how acquisition order, social usage, and instructional context collectively orchestrate cross-linguistic activation [19].

To probe these interacting factors, a methodology capable of capturing the distinct dynamics of both language production and comprehension is required. The present study therefore implements a production-to-comprehension priming paradigm—a design specifically chosen to bridge a significant gap in a literature dominated by comprehension-based tasks. We combine a picture naming task (speech production) during the study phase with a lexical decision task (visual word recognition) during the test phase. This approach directly probes how conceptual activation engaged during speech influences subsequent reading-related access, providing much-needed evidence to test and extend models like Multilink or BIA-d [9,20]. Crucially, building on models of context-dependent lexical activation [8], we operationalize ‘context’ as the ‘language pairing context’—the combination of languages used across the study and test phases (e.g., L1→L2 vs. L3→L2). This manipulation enables us to systematically investigate how two key factors modulate cross-linguistic activation: the age of acquisition of the prime language, linked to conceptual connection strength [5], and the social usage frequency of the target language, tied to its resting-level activation [9].

This experimental design directly translates our investigation of these modulating factors into two primary research questions. First (RQ1), do trilingual speakers exhibit cross-linguistic co-activation across all three of their languages? Consistent with non-selective access accounts, we expect to find significant cross-language priming effects. Our central research question is therefore (RQ2): How do factors of acquisition order and language usage frequency shape the asymmetrical patterns of this cross-linguistic priming?

To answer this central question, we formalize the potential roles of these factors into two competing hypotheses that make distinct, fine-grained predictions. The first, the Age of Acquisition (AoA) Primacy Hypothesis, is derived from developmental frameworks such as the Revised Hierarchical Model [4]. This hypothesis concerns the power of the prime language, positing that the strength of its conceptual links is determined by developmental history. It thereby predicts a monotonic decrease in priming strength from earlier- to later-acquired languages. Crucially, testing this prediction is only possible when the target language is held constant—a methodological advantage unique to trilingual research. Whereas traditional bilingual studies comparing L1→L2 and L2→L1 priming are confounded because both the prime and target languages differ across conditions, a trilingual design allows us to isolate the influence of the prime. For instance, when L2 is the target, the AoA hypothesis predicts that priming from L1 will be stronger than from L3 (i.e., L1→L2 > L3→L2). This pattern would reflect a more potent signal from the established L1 system.

Shifting the theoretical focus from the power of the prime to the receptivity of the target, the Social Usage Frequency Hypothesis, informed by usage-based models like Multilink [9], presents a competing account. We refine the concept of usage frequency by distinguishing between “home use” and “social use,” a core distinction identified by Anderson et al. [21]. This distinction is particularly relevant for our population, as their L1 (Uyghur) is a heritage language largely confined to the home. Crucially, its usage decreased significantly as participants entered the L2-Chinese educational and social environment. We therefore hypothesize that social usage frequency—driven by interactions in the wider community—is the primary modulator of a language’s resting-level activation. This account posits that the language with the highest social usage (L2-Chinese) will be most susceptible to priming. It therefore predicts that the largest priming effects will be observed when L2 is the target. For example, holding the priming language constant, the effect on L2 should be greater than on L3 (e.g., L1→L2 > L1→L3), because L2′s higher activation makes its lexical representations easier to access.

Taken together, these hypotheses create a direct test between two powerful, competing forces: the developmental history dictating the power of the prime versus the current social usage determining the receptivity of the target. By testing these specific predictions in our unique population, we can directly disentangle their respective contributions. This mapping of asymmetrical activation pathways will provide the foundational model needed to understand how the brain orchestrates the flow of information from speech production to reading comprehension.

## 2. Materials and Methods

### 2.1. Research Objectives

To bridge a critical gap in our understanding of language architecture, this study investigates the functional interaction between the motor act of speech production and the perceptual-cognitive process of reading comprehension. Using a unique trilingual population, we aim to establish a fine-grained baseline model of cross-linguistic activation. Specifically, our objectives are to test for cross-modal priming from speech production to reading comprehension across all three languages of a trilingual speaker, thereby providing evidence for the integrated nature of the multilingual lexicon. Furthermore, we will systematically examine how the magnitude of this production-to-comprehension priming is regulated by the specific pairing of the languages used, offering insight into the cognitive control mechanisms that govern these cross-system interactions.

### 2.2. Methods

#### 2.2.1. Participants

One hundred and forty-four undergraduates participated in Experiment with monetary compensation. Ages ranged from 17 to 24 years (*M* = 20.6, *SD* = 2.25). The study received ethical approval from South China Normal University’s IRB (IORG0011738). All participants provided written informed consent prior to participation. They were informed of their right to withdraw from the study at any time without penalty.

Inclusion criteria comprised language background (no fourth language acquisition, no immigration background), sensory function (normal or corrected-to-normal vision), and handedness (right-handed only). The exclusion of individuals with fourth-language acquisition was applied to maintain a controlled experimental design, ensuring that the observed effects were solely attributed to the trilingual (L1–L2–L3) system without interference from additional multilingual experience.

Following Brysbaert [22], we set the minimum meaningful effect size for the main effect of Production Language at Cohen’s *f* = 0.20. Power analysis using G*Power 3.1 [23] for a mixed ANOVA (*f* = 0.2, *α* = 0.05, power = 0.8, correlation = 0.5, 2 within-subject measurements, 6 between-subjects groups) indicated a required sample size of 96 participants. We recruited 144 participants to ensure robust statistical power.

Participants completed the Language and Social Background Questionnaire (LSBQ; [21]). They rated L1, L2, and L3 proficiency across four domains (speaking, comprehension, reading, writing) on 10-point scales. They also rated usage frequency on 5-point scales (see Table 1).

##### Language Acquisition and Proficiency Patterns

In this study, “Age of acquisition” was defined as the age at which participants first began systematic exposure to and learning of each language. A repeated measures ANOVA revealed significant differences in the age of acquisition across the three languages, *F*(2, 286) = 722, *p* < 0.001, *η*^2^ = 0.74. Specifically, Uyghur (L1) was acquired earliest (*M* = 1.24 years), followed significantly later by Chinese (L2) (*M* = 6.14 years; *t*(143) = 16.7, *p* < 0.001, Cohen’s *d* = 0.81). English (L3) was acquired last (*M* = 12.34 years; *t*(143) = 37.9, *p* < 0.001, Cohen’s *d* = 1.10).

To further contextualize the participants’ linguistic background, we examined “Residential Exposure,” defined as the duration of residence in communities where a specific language is primarily spoken. For English (L3), a value of 0 confirms that no participant had resided in Anglophone countries (e.g., the UK or USA), despite receiving formal classroom instruction. A repeated measures ANOVA revealed significant disparities in exposure duration across the three languages, *F*(2, 286) = 639, *p* < 0.001, *η*^2^ = 0.749. Participants reported the longest residential exposure to Chinese (*M* = 16.8 years), followed by Uyghur (*M* = 13.9 years), and English (*M* = 0 years). Post hoc pairwise comparisons confirmed that exposure to Chinese was significantly longer than to Uyghur, *t*(143) = 5.69, *p* < 0.001, Cohen’s *d* = 0.67. Furthermore, exposure to both Chinese and Uyghur significantly exceeded that of English (*t*(143) = 33.40, *p* < 0.001, Cohen’s *d* = 3.94; and *t*(143) = 27.71, *p* < 0.001, Cohen’s *d* = 3.27, respectively).

While residential data highlighted the environmental dominance of Chinese, the analysis of “Duration of Use” revealed a different hierarchy rooted in the participants’ acquisition history. This metric was calculated by subtracting the age of acquisition from the current age, excluding periods of non-use. A repeated measures ANOVA revealed significant differences in language usage history across the three languages, *F*(2, 286) = 418, *p* < 0.001, *η*^2^ = 0.66. Reflecting their status as native speakers, participants possessed the longest history of using Uyghur (*M* = 19.0 years), followed by Chinese (*M* = 14.1 years), and English (*M* = 8.1 years). Post hoc pairwise comparisons confirmed that all three languages differed significantly from each other: Uyghur > Chinese (*t*(143) = 13.10, *p* < 0.001, Cohen’s *d* = 1.54), Uyghur > English (*t*(143) = 28.90, *p* < 0.001, Cohen’s *d* = 3.40), and Chinese > English (*t*(143) = 15.80, *p* < 0.001, Cohen’s *d* = 1.86).

To examine whether distinct usage histories translated into corresponding differences in current ability, we evaluated language proficiency across four key dimensions: speaking, listening, reading, and writing. The results revealed a complex pattern of dominance that varied by skill, though proficiency in the third language (L3) remained consistently and significantly lower than both L1 and L2 across all domains. Consistent with usage history, L1 retained dominance in both speaking and reading. Participants demonstrated significantly higher proficiency in L1 speaking (*M* = 9.51) compared to L2 (*M* = 9.13), *t*(143) = −3.59, *p* = 0.001. Similarly, L1 reading proficiency (*M* = 8.56) was significantly higher than L2 (*M* = 7.97), *t*(143) = −5.32, *p* < 0.001. In both domains, performance in L3 was significantly lower than L1 (Speaking: *M* = 5.54, *t*(143) = −37.46, *p* < 0.001; Reading: *M* = 5.92, *t*(143) = −23.80, *p* < 0.001) and L2 (Speaking: *t*(143) = −33.87, *p* < 0.001; Reading: *t*(143) = −18.48, *p* < 0.001). In contrast, this pattern of L1 dominance did not extend to listening, which showed functional convergence between the two primary languages. No significant difference was found between L1 (*M* = 9.08) and L2 (*M* = 8.89) listening proficiency, *t*(143) = −2.09, *p* = 0.113. Nevertheless, comparisons with the third language remained consistent, as L3 listening proficiency (*M* = 5.06) was significantly lower than both L1 (*t*(143) = −44.16, *p* < 0.001) and L2 (*t*(143) = −42.07, *p* < 0.001). Most notably, the writing dimension displayed a reverse dominance pattern, deviating from the historical usage hierarchy. L2 writing proficiency (*M* = 8.79) was significantly higher than L1 (*M* = 7.28), *t*(143) = 11.6, *p* < 0.001. Despite this reversal, both languages were still significantly more proficient than L3 (*M* = 5.60), as indicated by comparisons with L1 (*t*(143) = −12.9, *p* < 0.001) and L2 (*t*(143) = −124.4, *p* < 0.001). Finally, we compared composite scores averaged across all four dimensions to assess overall ability. The analysis indicated that the specific domain variations balanced each other out: overall, there was no significant difference between the aggregate proficiency of L1 (*M* = 8.6075) and L2 (*M* = 8.695), *t*(143) = 1.33, *p* = 0.554. Consistent with the individual skills, overall L3 proficiency (*M* = 5.53) remained significantly lower than both L1 (*t*(143) = −46.66, *p* < 0.001) and L2 (*t*(143) = −47.99, *p* < 0.001). We further explored subgroup differences based on individual dominance scores. This analysis confirmed that the vast majority of participants individually mirrored the aggregate pattern: displaying L1 dominance in speaking and reading, equivalent proficiency in listening, and L2 dominance in writing. Thus, while objective usage history favors L1, the functional proficiency data indicate that L2 capabilities have largely converged with L1 levels.

To validate these subjective patterns and rigorously evaluate the actual lexical accessibility resulting from this acquisition structure, we moved beyond subjective reports. Addressing the potential limitations of self-ratings, which can vary across cultures [24], we administered the Multilingual Naming Test (MINT; [25]). The MINT is a standardized 68-item picture-naming measure that uses culturally neutral, high-frequency images to assess lexical retrieval abilities in multilingual individuals. Participants name each item as quickly and accurately as possible, with only conventional target-language names accepted (no partial credit). Scores reflect the number of correct responses (maximum = 68). Crucially, the objective data corroborated the functional convergence observed in the self-reports. MINT scores revealed no significant difference between Uyghur (*M* = 65.0) and Chinese (*M* = 64.8), *t*(143) = 0.634, *p* = 0.527, Cohen’s *d* = 0.074. However, reflecting the indirect acquisition of the L3, English scores were significantly lower than both Uyghur, *t*(143) = 47.87, *p* < 0.001, Cohen’s *d* = 0.553, and Chinese, *t*(143) = 47.24, *p* < 0.001, Cohen’s *d* = 0.479.

With the functional equivalence between L1 and L2 now confirmed by objective measures, we next sought to identify the sociolinguistic drivers that maintain this equilibrium. To elucidate the environmental factors sustaining this bilingual balance, we examined current usage patterns across two distinct domains: home and social contexts. Both “Frequency of Use: Home” and “Frequency of Use: Social” were rated on a 0–5 scale, where 0 indicates no usage and 5 indicates exclusive usage.

First, Home usage assessed the frequency and extent to which participants used each language within the family environment. This dimension encompassed communication with family members (parents, siblings, and spouse/partner) and language use during daily household activities such as mealtimes, leisure activities, and everyday conversations. Example questions included: “How often do you use Uyghur when speaking with your parents?” and “How often do you use Chinese when speaking with your siblings?” Results indicated that the home domain remains a stronghold for the native language. Home usage differed significantly across languages, *F*(2, 286) = 475, *p* < 0.001, *η*^2^ = 0.76. Uyghur was used most frequently at home (*M* = 4.11), followed by Chinese (*M* = 0.68), and English (*M* = 0.21). All pairwise comparisons were significant: Uyghur > Chinese, *t*(143) = 26.10, *p* < 0.001, Cohen’s *d* = 2.97; Uyghur > English, *t*(143) = 24.70, *p* < 0.001, Cohen’s *d* = 3.22; Chinese > English, *t*(143) = 10.60, *p* < 0.001, Cohen’s *d* = 1.25.

In sharp contrast, Social usage—which assessed the frequency of language use in contexts outside the home such as with friends, colleagues, and in public settings (school, workplace, social media)—revealed a reversal in dominance. Example questions included: “How often do you use Uyghur when socializing with friends?” and “How often do you use English at work?”. Social usage differed significantly across languages, *F*(2, 286) = 432, *p* < 0.001, *η*^2^ = 0.74. Chinese was used most frequently in social contexts (*M* = 4.29), followed by Uyghur (*M* = 0.66), and English (*M* = 0.00). All pairwise comparisons were significant: Chinese > Uyghur, *t*(143) = 25.20, *p* < 0.001, Cohen’s *d* = 2.68; Uyghur > English, *t*(143) = 11.70, *p* < 0.001, Cohen’s *d* = 1.38; Chinese > English, *t*(143) = 36.90, *p* < 0.001, Cohen’s *d* = 3.06.

To contextualize these distinct usage patterns within the participants’ educational trajectory, we further categorized their “Learning Contexts” and the “Medium-of-Instruction.” “Learning Contexts” define the primary environments of acquisition (home, school, or both). Consistent with the observed home-social dichotomy, participants learned their heritage L1 (Uyghur) exclusively at home, whereas the national L2 (Chinese) was acquired formally at school. Furthermore, the “Medium-of-Instruction”—referring to the language used for teaching in school—played a critical role in the trilingual hierarchy. In this study, L3 (English), a foreign language, was learned with L2 (Chinese) serving as the medium of instruction. This configuration implies a tiered acquisition structure where English was accessed indirectly through the participants’ second language rather than their native tongue.

#### 2.2.2. Materials

The core stimuli comprised forty-eight pictures sourced from the International Picture Naming Project database [26], each paired with its corresponding name in Uyghur, Chinese, and English (see Appendix B for sample stimuli). A critical prerequisite for their use was ensuring their conceptual and linguistic equivalence across these distinct languages, thereby validating them for the experiment. Accordingly, a norming study was conducted with twenty-five Uyghur–Chinese–English trilinguals (who did not participate in the main experiment) to assess these properties. Participants rated the familiarity of the vocabulary and the accuracy of picture-word pairings on a 7-point scale. Statistical analysis confirmed the materials were well-balanced, revealing no significant cross-linguistic differences in familiarity (*f*(2, 94) = 1.73, *p* = 0.181, *η*^2^ = 0.019), picture-word matching accuracy (*F*(2, 94) = 0.390, *p* = 0.677, *η*^2^ = 0.002), or syllable count (*F*(2, 94) = 3.04, *p* = 0.062, *η*^2^ = 0.013).

While the stimuli were balanced on these key metrics, a further analysis was necessary to quantify the inherent differences in their physical forms across scripts. To this end, we examined word length, measured by the number of letters for Uyghur and English and by the number of strokes for Chinese. A one-way repeated measures ANOVA revealed a highly significant main effect of language on this measure, *F*(2, 94) = 64.3, *p* < 0.001, *η*^2^ = 0.449. Post hoc tests specified that the mean length of Chinese words (*M* = 11.88, *SD* = 6.85) was significantly greater than that of both English words (*M* = 4.77, *SD* = 1.23; *t*(47) = 10.006, *p* < 0.001) and Uyghur words (*M* = 5.05, *SD* = 1.64; *t*(47) *=* −9.619, *p* < 0.001). However, no significant difference was found between the mean lengths of Uyghur and English words (*t*(47) = 0.388, *p* = 0.699).

A critical component of the experimental design was the creation of language-specific pseudowords that are pronounceable and orthographically plausible but semantically meaningless—to serve as novel foils in the experimental task. The construction process was tailored to the unique features of each script. For the alphabetic scripts of English and Uyghur, this involved a subsyllabic structure matching approach, where a single phonological segment of a real word was replaced to create a legal non-word (e.g., English “milk”→“misk”; Uyghur “گۈل” /ɡyl/→“گول” /ɡol/). In contrast, due to the logographic nature of Chinese, a two-tiered approach was adopted. This process involved either the creation of pseudo-characters from real radicals (e.g., combining components to form a novel character) or the combination of real characters into meaningless phrases (e.g., “架析” jià xī, a semantically anomalous phrase that literally translates to ‘shelf analysis’). To implement these methods, English pseudowords were generated using the Wuggy generator [27], resulting in an average OLD20 (Orthographic Levenshtein Distance to the 20 nearest neighbors) of 1.46 (SD = 0.31). In contrast, Uyghur pseudowords were created by modifying real words following vowel harmony rules, with a mean Levenshtein distance of 1.32 (SD = 0.42) relative to their corresponding base words. For Chinese, single-character pseudowords were constructed using the Chinese pseudo-character/non-character producing (CPN) system [28], while two-character pseudowords were manually created by recombining real characters. The full list of real words and generated pseudowords across all three languages is provided in Table A1 in Appendix A.

Given that the pseudowords were constructed using script-specific methods, it was crucial to validate their experimental function and control for potential confounds in our cross-linguistic comparison. We addressed this through a two-pronged approach targeting both stimulus validation and analytical strategy.

First, at the stimulus level, we ensured the materials were rigorously controlled. All pseudowords were initially validated for their unfamiliarity and non-word status through pretesting with 25 Uyghur–Chinese–English trilinguals who did not participate in the main study. Furthermore, to ensure comparability between real words and pseudowords, we conducted a series of statistical analyses. A 2 (Lexicality: real word vs. pseudo-word) × 3 (Language: Uyghur vs. Chinese vs. English) mixed-design ANOVA on the number of basic orthographic units (letters for Uyghur/English, strokes for Chinese) initially revealed a significant main effect of Language (*F*(2, 282) = 102.776, *p* < 0.001, *η*^2^ = 0.422), which was expected, as Chinese characters inherently contain more strokes than words in alphabetic scripts have letters. Critically, there was no significant main effect of Lexicality (*F*(1, 282) = 0.229, *p* = 0.632) and no Lexicality × Language interaction (*F*(2, 282) = 0.108, *p* = 0.898), demonstrating that real words and pseudowords were successfully matched on physical complexity within each language.

To create a standardized measure of orthographic complexity that was directly comparable across these different unit types (strokes vs. letters), we converted all length values to z-scores within each language. A subsequent ANOVA on these standardized scores confirmed the success of this method: the previously significant main effect of Language was eliminated (*F*(2, 282) = 0.0001, *p* = 0.998). Moreover, the main effect of Lexicality (*F*(1, 282) = 0.24, *p* = 0.625) and the Lexicality × Language interaction (*F*(2, 282) = 0.0255, *p* = 0.975) remained non-significant, providing robust evidence that stimulus length was equivalent across all conditions after standardization.

Second, at the analysis level, our primary strategy focused on within-language priming effects (i.e., the reaction time difference between learned and unlearned words), a relative measure that internally controls for baseline processing speed differences across languages. The success of this comprehensive approach, which rested on both rigorous stimulus control and this analytical strategy, was confirmed from two sides. Behaviorally, low error rates (≤4.34%) in the subsequent experiments indicated that participants could reliably distinguish the validated pseudowords from real words. Structurally, the final component of our stimulus matching—syllable counts—was also balanced. A 2 × 3 ANOVA on syllable counts confirmed no significant main effect for Lexicality, *F*(1, 282) = 0.229, *p* = 0.632, and no significant Lexicality × Language interaction, *F*(2, 282) = 0.00828, *p* = 0.928, confirming that real-word stimuli and pseudoword fillers were appropriately matched in their syllabic structure across all three writing systems.

#### 2.2.3. Design

The study employed a 2 (Lexical Status: studied vs. unstudied) × 2 (Production Language: earlier-acquired vs. later-acquired) × 3 (Recognition Language: L1, L2, L3) mixed design. To implement this design, a total of 144 participants were recruited and randomly assigned to one of the six between-subjects language conditions (n = 24 per condition). These conditions, determined by the pairing of the production and recognition languages, were: Uyghur→Chinese, Chinese→Uyghur, Chinese→English, English→Chinese, Uyghur→English, and English→Uyghur. Lexical Status served as the within-subjects factor. The primary dependent variable was reaction time (RT), with the cross-linguistic priming effect operationally defined as the RT difference between unstudied and studied items (RT_unstudied—RT_studied).

To operationalize the crucial within-subjects factor of Lexical Status, the experimental procedure was divided into two distinct phases: a production phase (picture naming) followed by a recognition phase (lexical decision task). The stimuli for these tasks consisted of 48 experimental pictures sourced from the International Picture Naming Project database [26]. To control for item-specific confounds, these pictures were first partitioned into two equivalent sets (Set A and Set B, n = 24 each), which were then used to create two master experimental lists (List 1 and List 2). Within each of the six language conditions, participants were randomly assigned to one of these lists (n = 12 per list), a step that determined which items would serve as ‘studied’ versus ‘unstudied’. For instance, in the Chinese→English condition, a participant assigned to List 1 would name a picture of an apple in Chinese (“苹果”), making the English word “apple” a ‘studied’ item. For a participant in List 2, “apple” would be an ‘unstudied’ item. This fully counterbalanced design ensures every item appeared equally often in both conditions, isolating any calculated priming effect from item-specific properties. To construct the final recognition phase, these 48 target words (24 studied, 24 unstudied) were presented alongside an equal number of 48 pronounceable, language-specific pseudowords (e.g., driss in English) that served as foils. The presentation of target words and pseudowords was randomly intermixed for each participant, as was the order of all trials within each phase.

#### 2.2.4. Procedure

The experimental procedure, programmed in E-Prime 2.0, was designed to measure the influence of motor speech production on subsequent visual word recognition (see Figure 1 for an example trial). Participants were unaware of the relationship between the experiment’s two distinct phases. The first was a production phase, where participants performed a vocal picture-naming task in a single, pre-assigned language. Each trial began with a 500 ms fixation cross, followed by a picture with a colored border indicating the required naming language. The picture remained visible for up to 5000 ms or until a vocal response was registered. Following a one-minute break, participants entered the recognition phase, where they performed a visual lexical decision task (LDT) in a different language. In each LDT trial, a fixation cross appeared for 500 ms, followed by a letter string. Participants made a speeded judgment (“real word” vs. “pseudoword”) via keypress. To ensure the priming effect was not confounded by response type, both studied items (translations of the named pictures) and unstudied items were real words. The resulting reaction time difference (RT unstudied—RT studied) served as our primary measure of cross-modal facilitation, quantifying the extent to which the initial motor act of production influenced the speed of subsequent perceptual recognition.

#### 2.2.5. Data Analysis

Data analysis was performed in R (Version 4.4.3) using linear mixed-effects models (LMM) via the lme4 [29] and lmerTest packages. Given that accuracy in the lexical decision task was at ceiling (>95%), the analysis focused exclusively on reaction times (RTs) for correct responses. Prior to modeling, we applied a data trimming procedure outlined by Chen [19], excluding trials with RTs exceeding ±3 standard deviations from each participant’s mean. We adopted this criterion to minimize the influence of extreme values representing attentional lapses or anticipatory motor errors [30], while satisfying the normality assumption of residual distributions required by linear mixed-effects models [31]. This cleaning process resulted in a total data loss of 4.34%.

Fixed Effects and Coding: To analyze the cleaned RT data, we fitted a linear mixed-effects model examining the effects of Lexical Status (studied vs. unstudied), Production Language (earlier- vs. later-acquired), Recognition Language (L1, L2, L3), and their interactions. Categorical predictors were contrast-coded to facilitate the interpretation of main effects and interactions. Specifically, Lexical Status and Production Language were sum-coded (0.5 vs. −0.5), ensuring that the main effects reflected the overall differences across conditions. For Recognition Language, we manually specified a contrast matrix to test specific research hypotheses via pairwise comparisons: the first contrast compared L2 vs. L1 (coded: L2 = 0.5, L1 = −0.5, L3 = 0); the second compared L3 vs. L1 (coded: L3 = 0.5, L1 = −0.5, L2 = 0); and the third compared L2 vs. L3 (coded: L2 = 0.5, L3 = −0.5, L1 = 0).

Random Effects Structure: To account for the non-independence of observations and to ensure the results generalize to the broader populations of participants and items, we adopted a maximal random effects structure as recommended by Barr et al. [32]. The final model included random intercepts for both participants (SubjectID) and items (ItemID). Crucially, to control for individual variation in the magnitude of the priming effect, we also included by-participant and by-item random slopes for the within-unit factor, Lexical Status. The final model formula was specified as: RT ~ Lexical_Status × Recognition_Language × Production_Language + (1 + Lexical_Status|SubjectID) + (1 + Lexical_Status|ItemID).

Significance Testing and Effect Sizes: Statistical significance for fixed effects was assessed using Satterthwaite’s method [33] to estimate degrees of freedom and calculate *p*-values, as implemented in the lmerTest package [34]. This approach was selected for its ability to produce acceptable Type I error rates for psycholinguistic data [35]. To quantify the magnitude of the observed effects, we calculated effect sizes (Cohen’s *d*) for fixed effects and planned contrasts using the emmeans package [36,37]. Specifically, *d* was computed as the difference between the estimated marginal means divided by the square root of the model’s residual variance (sigma) [38].

## 3. Results

The primary objective of our analysis was to empirically test the two competing hypotheses established in the Introduction: the Age of Acquisition Primacy Hypothesis (focusing on the power of the prime) and the Social Usage Frequency Hypothesis (focusing on the receptivity of the target). By leveraging the production-to-comprehension paradigm, we examined how the specific pairing of Production Language (Prime) and Recognition Language (Target) modulates the magnitude of repetition priming.

We present the findings in two stages to directly address our research questions. First, we evaluate the fundamental existence of cross-linguistic co-activation (RQ1) by examining the overall effect of Lexical Status. Second, and most critically, we resolve the theoretical tension regarding asymmetry (RQ2) by dissecting the significant interactions between Production and Recognition languages. This allows us to determine whether the priming magnitude is driven more by the developmental history of the prime or the social usage frequency of the target.

Descriptive statistics for the reaction times (RTs) are visualized in Figure 2, and the specific cross-language repetition priming effects (the magnitude of the difference between unstudied and studied words) are detailed in Table 2 and Figure 3. The complete statistical parameters from the linear mixed-effects model are provided in Table 3.

### 3.1. Evidence for Cross-Modal Facilitation (Addressing RQ1)

To address the first research question regarding the functional link between speech production and word recognition, we examined the main effect of Lexical Status. The analysis revealed a significant and reliable cross-language priming effect: responses to studied words (*M* = 863 ms) were significantly faster than to unstudied words (*M* = 1012 ms), *t*(126) = 35.5, *p* < 0.001, Cohen’s *d* = 0.431. This result provides the foundational evidence for cross-modal facilitation, confirming that the act of naming a picture systematically enhances the efficiency of recognizing its written translation.

### 3.2. Baseline Efficiencies of Production and Recognition

Before examining the interactions, we assessed the independent characteristics of the production and recognition systems.

First, regarding Production Language, naming in an earlier-acquired language led to faster subsequent recognition overall (*M* = 893 ms) compared to a later-acquired language (*M* = 982 ms), *t*(131) = 6.00, *p* < 0.001, Cohen’s *d* = 0.202.

Second, regarding Recognition Language, a significant effect was observed (*ps* < 0.001). Response times followed a proficiency-based pattern: L1 (888 ms) ≈ L2 (885 ms) < L3 (1040 ms).

### 3.3. Drivers of Asymmetry: Testing the Two Hypotheses (Addressing RQ2)

Production-Driven Asymmetry: A significant interaction between Production Language and Lexical Status was found, *t*(126) = 5.72, *p* < 0.001, Cohen’s *d* = 0.401. Specifically, naming in an earlier-acquired language generated a larger priming effect (173 ms) compared to a later-acquired language (125 ms). This pattern aligns with the Age of Acquisition Primacy Hypothesis.

Recognition-Driven Asymmetry: A significant interaction between Recognition Language and Lexical Status was also observed. Pairwise comparisons indicated that the magnitude of priming differed across target languages: L2 (175 ms) > L1 (156 ms) > L3 (114 ms) (all *ps* ≤ 0.010). This result supports the Social Usage Frequency Hypothesis, as the most frequently used L2 benefited most from the prime.

## 4. Discussion

### 4.1. The Integrated and Asymmetrical Architecture of the Trilingual Lexicon (Addressing RQ1)

The primary purpose of this study was to elucidate the functional architecture connecting the motor system of speech production with the perceptual-cognitive system of reading comprehension in the trilingual brain. Specifically, we aimed to determine whether the motor act of naming a picture facilitates the subsequent recognition of its translation equivalents. Our results provided a clear affirmative answer: we observed a significant cross-language priming effect across languages. This finding confirms that the trilingual lexicon acts as a single, integrated system where all languages remain co-activated [39,40,41].

However, this co-activation is not uniform. Our central finding reveals a fundamental functional dissociation. The top-down influence flowing from production-to-comprehension is consistently driven by Age of Acquisition (AoA). This pattern reflects the unique strength of concept-to-lexicon links forged in early life. In contrast, the bottom-up efficiency of the recognition system itself is governed by social usage frequency, reflecting the dynamic resting-state activation of lexical representations. This dissociation suggests that cross-linguistic interactions are not symmetrical but are asymmetrically weighted [42,43]. Ultimately, by mapping these interaction patterns, our findings highlight that the trilingual lexicon is not a monolithic entity. Instead, it is a complex system that continuously balances stable historical constraints (AoA) with dynamic environmental demands (social usage).

### 4.2. Resolving the Theoretical Tension: AoA Primacy vs. Social Usage (Addressing RQ2)

Our central research question (RQ2) sought to resolve the tension between two competing theoretical accounts: the AoA Primacy Hypothesis (power of the prime) and the Social Usage Frequency Hypothesis (receptivity of the target). Our data indicate that these are not mutually exclusive alternatives, but rather distinct mechanisms governing different ends of the processing chain.

#### 4.2.1. The Power of the Prime: Support for AoA Primacy

Regarding the top-down influence originating from speech production, our data indicate that AoA is the primary determinant. Activating an earlier-acquired language (L1 or L2) during picture naming generated significantly larger cross-language priming effects than using the later-acquired L3.

This pattern aligns precisely with developmental models such as the Revised Hierarchical Model (RHM), which posit that early language learning establishes more robust conceptual-lexical links [4,5,44]. In our task, picture naming requires retrieving a lemma directly from a non-linguistic concept. The finding that early-acquired languages produce a consistent facilitation suggests that they trigger a more effective spreading of activation from the conceptual level. This interpretation is consistent with prior neurocognitive research indicating that L1 processing elicits deeper semantic engagement than formal L2/L3 instruction [45]. Thus, the “power” of the top-down signal is a function of acquisition history: early-forged links create a reliable signal that propagates effectively to other languages.

Crucially, this top-down facilitation was global—using a dominant language facilitated, rather than inhibited, the recognition of other languages.

#### 4.2.2. The Receptivity of the Target: Support for Social Usage Frequency

In contrast, the bottom-up recognition system followed a different set of principles governed by social usage patterns. While baseline reaction times reflected standard proficiency effects (L1/L2 > L3) consistent with established recognition thresholds [46,47], the magnitude of the priming benefit revealed a specific hierarchy: L2 Chinese (socially dominant) showed the largest priming effect, followed by L1 Uyghur, with L3 English showing the least.

This pattern (L2 > L1 > L3) challenges the assumption that “frequency of use” is a monolithic variable. We observed that Home Usage (highest for L1) did not predict the priming hierarchy, whereas Social Usage (highest for L2) mirrored the observed priming magnitude. This distinction draws upon the functional specialization of usage domains explored by Anderson et al. [21].

Why does Social Usage trump Home Usage? We propose that this disparity reflects the ‘tuning’ of resting-level activation thresholds based on contextual demands. Home usage, while frequent, is often repetitive and predictable. In contrast, Social Usage involves interacting with diverse speakers and topics in a noisy environment. While our results support the Multilink model’s [9] core premise that usage frequency drives resting-level activation, our findings offer a crucial refinement. We demonstrate that Social Usage is the primary driver. This suggests that the contextual diversity inherent in social interaction is more effective at raising activation thresholds than the predictable context of home usage [48,49,50]. Consequently, L2 words act as more sensitive “receivers.” When the signal arrives from the prime, the highly activated L2 representations reach recognition thresholds faster. While the global co-activation aligns with frameworks like the BLINCS model [51], the distinctive interplay of factors underscores the urgent need for theoretical frameworks to move beyond direct applications of bilingual models to fully capture the unique complexities of the trilingual experience [52,53,54,55].

### 4.3. Methodological Contributions: The Utility of the Trilingual Design

This study makes a direct contribution to contemporary psycholinguistics by demonstrating the unique methodological value of trilingualism. In traditional bilingual research, untangling the effects of AoA from Usage is notoriously difficult. This is because switching the translation direction (e.g., L1→L2 vs. L2→L1) inevitably changes both the source (prime) and the target (recognition) languages simultaneously. This makes it impossible to determine whether an asymmetry is due to the “stronger push” of the prime or the “easier reception” of the target.

Our study overcomes this by utilizing the “triangular” nature of trilingualism. By employing a three-language design, we could hold one side of the equation constant while varying the other. For instance, by comparing conditions such as L1→L2 and L1→L3, we were able to hold the production language (AoA of the prime) constant while varying the comprehension language (Social Usage of the target). This design effectively disentangles the contribution of the speech motor system from that of the reading system—a level of granular control unattainable in binary language models.

Furthermore, our approach addresses the question of the interaction between production and comprehension systems. We demonstrate that the information flow is not a uniform process but a context-sensitive mechanism. It dynamically recruits different functional components—historical conceptual links versus current lexical activation—to meet the distinct demands of speaking and reading. This context-sensitive mechanism validates the argument that trilingualism is not merely “bilingualism plus one,” but a distinct experimental state that offers a clearer window into the component processes of language control.

### 4.4. Limitations and Future Directions

The present study has several limitations that suggest avenues for future research. First, our findings are based on a specific trilingual population (Uyghur–Chinese–English) with a distinct split between home and social usage domains. While this split was methodologically advantageous, future work should examine populations with more balanced use to better disentangle the effects of usage patterns from AoA and proficiency. Second, the between-subjects design is susceptible to individual differences; a within-subjects design would offer more precise control to directly test variables like word frequency. Finally, a key challenge is the inherent visual processing asymmetry across scripts (e.g., logographic Chinese vs. alphabetic Uyghur/English). While we mitigated this by using relative priming effects as our main metric, this constraint remains. Future studies could more precisely calibrate cross-linguistic comparisons by establishing script-specific processing baselines, thereby enhancing the validity of the findings.

## 5. Conclusions

In conclusion, our study delves into the intricate interactions between speech production and reading comprehension within the trilingual cognitive architecture, revealing a fundamental asymmetry. We establish that the system operates via a dual mechanism. On the one hand, production is history-driven: it operates as a top-down process governed by the AoA Primacy principle. Here, the privileged conceptual access of the earliest-acquired language generates a distinct signal. On the other hand, recognition is environment-driven: it functions as a bottom-up process governed by the Social Usage Frequency principle. In this system, the demands of social interaction determine the resting-level receptivity of the lexicon.

## Figures and Tables

**Figure 1 brainsci-15-01288-f001:**
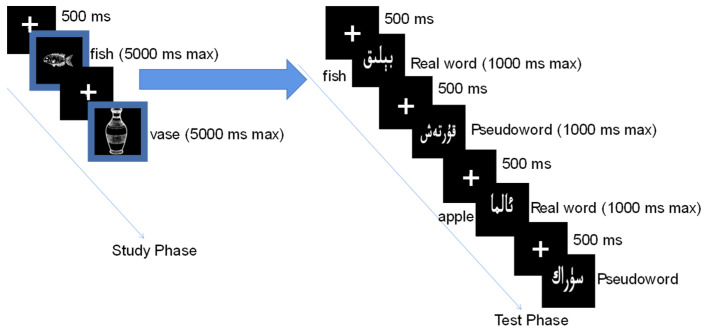
Schematic illustration of the experimental procedure. Note: In the Study Phase, the blue borders indicate that participants were required to name the pictures (e.g., ‘fish’ and ‘vase’) in English. In the Test Phase, the stimuli displayed in the black boxes are in Uyghur script. The item labeled ‘Real word’ (top) is the translation equivalent of the studied item (‘fish’), whereas the item labeled ‘Real word’ (bottom) is the translation for ‘apple’, which was not presented in the Study Phase. The items labeled ‘Pseudoword’ are pronounceable but meaningless character strings constructed to resemble real words.

**Figure 2 brainsci-15-01288-f002:**
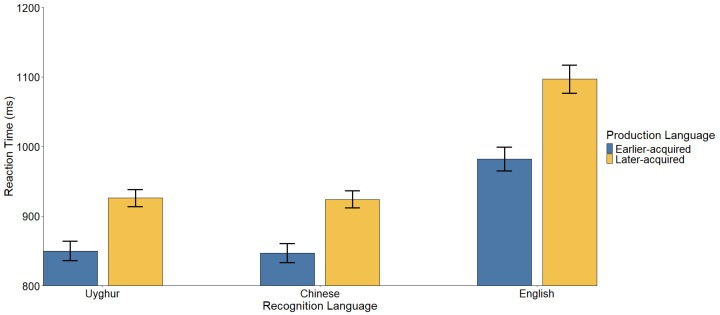
Mean reaction times (ms) in the lexical decision task. The x-axis represents the Recognition Language. The colored bars distinguish whether the preceding Production Language was acquired earlier (blue) or later (yellow). Error bars represent the standard error of the mean.

**Figure 3 brainsci-15-01288-f003:**
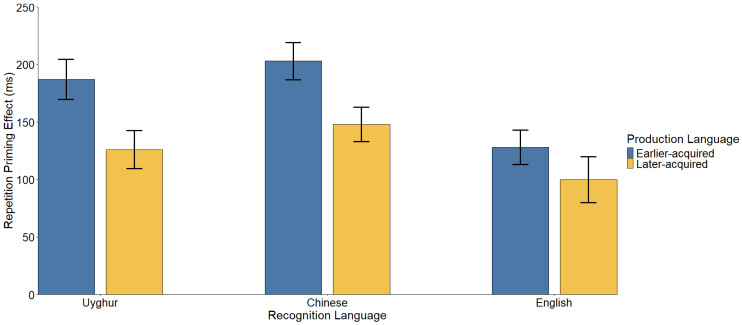
Cross-language repetition priming effects (ms). The x-axis represents the Recognition Language. The colored bars distinguish the magnitude of the priming effect generated when the preceding Production Language was acquired earlier (blue) or later (yellow). Error bars represent the standard error of the priming effect.

**Table 1 brainsci-15-01288-t001:** Participant Language Background.

Characteristic	L1 Uyghur	L2 Chinese	L3 English
Age of acquisition (years) *M* (*SD*)	1.24 (1.81)	6.14 (3.27)	12.34 (3.41)
Residential Exposure (years) *M* (*SD*)	13.9 (2.29)	16.8 (2.16)	0 (0)
Duration of Use (years) *M* (*SD*)	19.02 (2.61)	13.94 (3.28)	8.00 (3.76)
Frequency of Use (Home; 0–5) *M* (*SD*)	4.10 (0.47)	0.68 (0.44)	0.20 (0.41)
Frequency of Use (Social; 0–5) *M* (*SD*)	0.69 (0.29)	4.29 (0.46)	0.02 (0.40)
Self-ratings of proficiency			
Speaking *M* (*SD*)	9.51 (1.23)	9.13 (1.19)	5.54 (1.61)
Listening *M* (*SD*)	9.08 (1.21)	8.89 (1.06)	5.06 (1.20)
Reading *M* (*SD*)	8.56 (1.11)	7.97 (1.69)	5.92 (1.58)
Writing *M* (*SD*)	7.28 (1.12)	8.79 (1.99)	5.60 (1.87)
MINT (0–68) score *M* (*SD*)	65.0 (1.92)	64.8 (2.10)	49.8 (2.70)
Learning contexts			
Home-only Learning *N* (%)	88 (61.11%)	0 (0%)	0 (0%)
School-only Learning *N* (%)	0 (0%)	122 (84.72%)	144 (100%)
both *N* (%)	56 (38.89%)	22 (15.27%)	0 (0%)
Medium-of-instruction			
Uyghur *N* (%)	n/a	0 (0%)	0 (0%)
Chinese *N* (%)	n/a	144 (100%)	144 (100%)

Note. Values are M (SD). Frequency of Use scores are six-point Likert-type ratings (0–5; 0 = never … 5 = exclusively/always). Home = within-family interactions; Social = interactions outside the home (friends, school/work, public). Residential Exposure = years living in regions where the language is the community language (English = 0 indicates no residence in English-speaking countries). Duration of Use = current age − age of acquisition, excluding self-reported non-use periods. MINT = Multilingual Naming Test, score range 0–68.

**Table 2 brainsci-15-01288-t002:** Cross-Language Repetition Priming Effects (ms) as a Function of Recognition Language and Production Language.

	Uyghur	Chinese	English
Earlier-acquired language			
Unstudied words *M* (*SE*)	944 (11.26)	948 (10.46)	1046 (10.58)
Studied words *M* (*SE*)	757 (12.36)	745 (11.46)	918 (10.29)
Repetition priming effect *M* (*SE*)	187 (17.5)	203 (16.2)	128 (14.96)
Later-acquired language			
Unstudied words *M* (*SE*)	989 (12.02)	998 (11.28)	1147 (14.12)
Studied words *M* (*SE*)	863 (11.62)	850 (10.58)	1047 (13.06)
Repetition priming effect *M* (*SE*)	126 (16.44)	148 (14.96)	100 (19.98)

Note. The columns represent the Recognition Language (the language of the lexical decision task). The data are grouped into rows by the Production Language used during picture naming. *M* = mean; *SE* = standard error.

**Table 3 brainsci-15-01288-t003:** Summary of Fixed Effects for the Linear Mixed-Effects Model Predicting Reaction Times.

Fixed Effects	Estimate	*SE*	*df*	*t*	*p*
(Intercept)	928.2	7.41	141	126.52	<0.001
**Main Effects**					
**Lexical Status (Studied vs. Unstudied)**	**−148.6**	**4.42**	**126**	**−35.5**	**<0.001**
**Production Language (Earlier vs. Later)**	**89.18**	**14.83**	**131**	**6**	**<0.001**
Recognition Language (L2 vs. L1)	−2.99	4.54	138	−0.897	0.37
**Recognition Language (L3 vs. L1)**	**151.68**	**16.14**	**137**	**9.33**	**<0.001**
**Critical Interactions**					
**Lexical Status × Production Lang (Early vs. Late)**	**48.04**	**8.84**	**126**	**5.72**	**<0.001**
**Lexical Status × Recognition Lang (L2 vs. L1)**	**−18.65**	**7.82**	**136**	**−2.573**	**0.022**
**Lexical Status × Recognition Lang (L3 vs. L1)**	**42.3**	**10.47**	**138**	**4.73**	**<0.001**

Note. Significant effects are indicated in bold. L1 = Uyghur; L2 = Chinese; L3 = English. For the complete list of model parameters, including all pairwise comparisons and non-significant higher-order interactions, please refer to Appendix C, Table A3.

## Data Availability

The data presented in this study are available on request from the corresponding author, due to privacy restrictions.

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
