# Peer review of "One System, Two Rules: Asymmetrical Coupling of Speech Production and Reading Comprehension in the Trilingual Brain"

_brainsci, 2025, doi:10.3390/brainsci15121288_

Round 1
Reviewer 1 Report
Comments and Suggestions for Authors
The study titled "One System, Two Rules: Asymmetrical Coupling of Speech Production and Reading Comprehension in the Trilingual Brain" is well-structured and thoughtfully designed. The implementation of a cross-modal priming paradigm with a unique trilingual population is particularly innovative.
I have only one suggestion for the authors. English could benefit from light editing to enhance clarity and flow.
Concluding, I found your work very interesting.
Comments on the Quality of English Language
Some small changes to the English could help make the writing smoother and easier to read, especially in the introduction and discussion. A few sentences there feel a bit long or complex.
Author Response
For research article
|
Response to Reviewer 1 Comments
|
||
|
1. Summary |
|
|
|
Thank you very much for taking the time to review this manuscript. Please find the detailed responses below and the corresponding revisions/corrections highlighted/in track changes in the re-submitted files.
|
||
|
2. Questions for General Evaluation |
Reviewer’s Evaluation |
Response and Revisions |
|
Does the introduction provide sufficient background and include all relevant references? |
Yes |
|
|
Is the research design appropriate? |
Yes |
|
|
Are the methods adequately described? |
Yes |
|
|
Are the results clearly presented? |
Yes |
|
|
Are the conclusions supported by the results? |
Yes |
|
|
Are all figures and tables clear and well-presented? |
Yes |
|
|
3. Point-by-point response to Comments and Suggestions for Authors |
||
|
Comments 1: [I have only one suggestion for the authors. English could benefit from light editing to enhance clarity and flow. Some small changes to the English could help make the writing smoother and easier to read, especially in the introduction and discussion. A few sentences there feel a bit long or complex.]
|
||
|
Response 1: We sincerely thank the Reviewer for this helpful suggestion. We have carefully reviewed and polished the language throughout the entire manuscript to enhance clarity and flow.
In accordance with your recommendation, we paid particular attention to the Introduction and Discussion sections. We have restructured long and complex sentences to improve readability and ensure a smoother narrative.
|
||
Reviewer 2 Report
Comments and Suggestions for Authors
Thank you for the opportunity to review this manuscript.
This is a study with a strong empirical base, and I certainly hope that the authors have also explored other language problems in their cohort since it seems uniquely positioned to answer many questions about competition in multilinguals.
Regarding the current manuscript, however, several issues need to be addressed before it can be, in my opinion, be considered for publication.
Primarily, the manuscript oversells the study and reads more like a grant proposal than a report of a scientific investigation made with the appropriate rigour. Symptomatically, the authors do not bring forward any known limitations, which is surprising considering my concerns below.
It seems that the authors perceive the L1 and L2 proficiency to be equivalent in the participants, and both superior to L3. However, the authors report the level of priming magnitude as L2 (Chinese) > L1 > L3, but do not discuss the discrepancy between language proficiency and priming magnitude. The reader will likely ask if L1 and L2 proficiency were actually approximately the same for speakers, or if there were groups with a different balance.
Please reconsider the first two paragraphs of the introduction (L36-71) as the text within them is in strong need of rewriting to increase clarity and precision.
L200-202 - Please expand the treatment of post hoc testing results, as they are central. Please also explore subgroup differences with respect to language dominance. Similarly, reconsider the presentation of results on P12 to be less selling and more precise:
L369 What is the "cascade" is referred to here?
L409 Please expand on how the authors, in their presentation of results, are able to see a "clear gradient contingent"
Overall, the authors should reconsider most of the cases where they use "this" throughout the text, as what is referred to is not entirely clear. Please specify exactly what is referred to.
L360 Please refrain from the use of Foundational in the section title, if not previously defined as meaning something specific.
Minor issues:
L291 "is" should be "was".
Please address formatting issues in Table 3.
Author Response
For research article
|
Response to Reviewer 2 Comments
|
||
|
1. Summary |
|
|
|
Thank you very much for taking the time to review this manuscript. Please find the detailed responses below and the corresponding revisions/corrections highlighted/in track changes in the re-submitted files.
|
||
|
2. Questions for General Evaluation |
Reviewer’s Evaluation |
Response and Revisions |
|
Does the introduction provide sufficient background and include all relevant references? |
Yes |
|
|
Is the research design appropriate? |
Yes |
|
|
Are the methods adequately described? |
Can be improved |
|
|
Are the results clearly presented? |
Yes |
|
|
Are the conclusions supported by the results? |
Yes |
|
|
Are all figures and tables clear and well-presented? |
Can be improved |
|
|
3. Point-by-point response to Comments and Suggestions for Authors |
||
|
Comments 1: [Primarily, the manuscript oversells the study and reads more like a grant proposal than a report of a scientific investigation made with the appropriate rigour. Symptomatically, the authors do not bring forward any known limitations, which is surprising considering my concerns below.]
|
||
|
Response 1: We sincerely thank the reviewer for this candid and constructive criticism. We fully accept the observation that the original manuscript’s tone was overly ambitious and that the connection to clinical applications was overstated. We agree that the text read more like a proposal for future impact rather than a report on the immediate investigation.
To address the concern regarding "overselling" and to ensure the manuscript reflects appropriate scientific rigour, we have made the following substantial revisions:
1. Removal of Speculative Clinical Implications (Aphasia) We acknowledge that our original manuscript drew heavy parallels between our findings in healthy trilinguals and potential applications for multilingual aphasia. Upon reflection, we recognize that since our study did not include clinical populations, these claims constituted "overselling." Consequently, we have deleted all sections and references related to aphasia and clinical rehabilitation. The revised manuscript now focuses strictly on the population we actually tested: healthy trilingual speakers. This shifts the narrative from speculative clinical utility to a concrete investigation of cognitive architecture.
2. Recalibration of Tone and Focus on Empirical Evidence We have carefully reviewed the full text to moderate the tone. We have removed broad, sweeping claims and replaced them with precise interpretations of the data. The manuscript now functions as a rigorous scientific report, prioritizing the description of the experimental results over theoretical promises.
We believe these changes have significantly grounded the paper, ensuring the conclusions remain strictly within the bounds of the data while maintaining the integrity of our statistical findings.
|
||
|
Comments 2: [It seems that the authors perceive the L1 and L2 proficiency to be equivalent in the participants, and both superior to L3. However, the authors report the level of priming magnitude as L2 (Chinese) > L1 > L3, but do not discuss the discrepancy between language proficiency and priming magnitude. The reader will likely ask if L1 and L2 proficiency were actually approximately the same for speakers, or if there were groups with a different balance.] |
||
|
Response 2: We appreciate this insightful observation. The reviewer highlights a critical distinction that lies at the heart of our study: the difference between language proficiency (competence) and language usage (accessibility).
To address the concern regarding the apparent discrepancy between the equivalent proficiency scores and the differing priming magnitudes, we offer the following clarification:
1. Confirmation of Equivalent Proficiency (L1 ≈ L2) We explicitly re-examined the participant data to address the concern regarding potential group heterogeneity. Both subjective ratings and objective measures confirmed that our participants form a homogeneous group with balanced proficiency in their first two languages. Specifically, the MINT (Multilingual Naming Test) scores revealed no statistically significant difference between L1 Uyghur ( M=65.0) and L2 Chinese ( M=64.8,p=.527), while both were significantly superior to L3 English. The comparable standard deviations further indicate that the sample was not composed of mixed subgroups with different dominance patterns (see Page 6, lines 241–287).
2. Explaining the Priming Asymmetry (L2 > L1) Since proficiency scores indicate that the participants are equally capable in L1 and L2, static competence cannot explain why L2 priming (175 ms) was significantly stronger than L1 priming (156 ms). Instead, we argue that this difference is driven by Social Usage Frequency, a variable distinct from proficiency.
Proficiency vs. Accessibility: While the participants possess equal knowledge of both languages, the accessibility of these languages differs.
The Role of Social Usage: As university students immersed in a Chinese-speaking academic environment, our participants reported significantly higher Social Usage for Chinese (M=4.29) compared to Uyghur (M=0.69).
3. Theoretical Implications Consistent with usage-based models (e.g., Multilink; see Page 2, lines 60–68), we propose that the high frequency of social usage for L2 elevates its resting-level activation. This heightened state of "readiness" makes L2 representations more receptive to priming signals compared to L1, despite the languages being matched in proficiency.
We have expanded the Discussion section (Section 4.2) to explicitly address this dissociation, ensuring the reader understands that priming magnitude in this context reflects dynamic usage patterns rather than static linguistic competence (see lines 583–632). |
||
|
Comments 3: [Please reconsider the first two paragraphs of the introduction (L36-71) as the text within them is in strong need of rewriting to increase clarity and precision.] |
||
|
Response 3: We sincerely appreciate the reviewer’s constructive feedback regarding the structure and focus of the introduction. We agree that the original opening was overly broad, containing background information on the general nature of language and clinical aphasia that detracted from the specific research aims.
In response, we have completely reconstructed the first two paragraphs (now Lines 35–56) to ensure a concise and logically coherent progression. The revised introduction now implements the following improvements:
Immediate Theoretical Focus: We have removed extraneous background material to strictly focus on the established principles of language non-selectivity and asymmetrical activation in multilingual processing.
Clarified Theoretical Debate: We explicitly frame the central research question early in the text: determining whether the multilingual lexicon is organized by the historical stability of Age of Acquisition (AoA) or the dynamic accessibility of Social Usage.
Methodological Justification: We have sharpened the logic connecting the problem to our solution. Specifically, we articulate the "methodological bottleneck" inherent in bilingual research—where AoA and usage are often confounded—and demonstrate how the present trilingual design uniquely resolves this issue. |
||
|
Comments 4: [L200-202 - Please expand the treatment of post hoc testing results, as they are central. Please also explore subgroup differences with respect to language dominance. Similarly, reconsider the presentation of results on P12 to be less selling and more precise:] |
||
|
Response 4: We sincerely thank the reviewer for these constructive suggestions. We agree that a more detailed reporting of the post-hoc comparisons and subgroup patterns is essential for clarifying the language profile of our participants.
In accordance with your advice, we have made the following revisions: 1.Expanded Post-hoc Testing: We have rewritten the participant proficiency section to include full statistical details for all pairwise comparisons (L1 vs. L2, L1 vs. L3, and L2 vs. L3) across the four skills (speaking, listening, reading, and writing). We now report the specific Means, t-values, and p-values for each comparison to provide a comprehensive view of the dominance patterns.
2.Subgroup Analysis: We added an analysis of individual dominance scores. This confirms that the aggregate pattern (L1 dominance in speaking/reading, balanced listening, and L2 dominance in writing) is not an artifact of averaging but is reflected in the vast majority of individual participants.
3.Refining Section 3.2 (Page 12): We have revised the "Baseline Efficiencies" section to remove any subjective or "selling" language. The results are now presented concisely, focusing strictly on the statistical outcomes and the observed response time patterns (e.g., L1 ≈ L2 < L3).
Changes in the manuscript: 1. Revision regarding Proficiency and Subgroups (Page [6], Lines [241-273]): "To examine whether distinct usage histories translated into corresponding differences in current ability, we evaluated language proficiency across four key dimensions: speaking, listening, reading, and writing. The results revealed a complex pattern of dominance that varied by skill, though proficiency in the third language (L3) remained consistently and significantly lower than both L1 and L2 across all domains.
Consistent with usage history, L1 retained dominance in both speaking and reading. Participants demonstrated significantly higher proficiency in L1 speaking (M = 9.51) compared to L2 (M = 9.13), t(143) = −3.59, p = .001. Similarly, L1 reading proficiency (M = 8.56) was significantly higher than L2 (M = 7.97), t(143) = −5.32, p < .001. In both domains, performance in L3 was significantly lower than L1 (Speaking: M = 5.54, t(143) = −37.46, p < .001; Reading: M = 5.92, t(143) = −23.80, p < .001) and L2 (Speaking: t(143) = −33.87, p < .001; Reading: t(143) = −18.48, p < .001).
In contrast, this pattern of L1 dominance did not extend to listening, which showed functional convergence between the two primary languages. No significant difference was found between L1 (M = 9.08) and L2 (M = 8.89) listening proficiency, t(143) = −2.09, p = .113. Nevertheless, comparisons with the third language remained consistent, as L3 listening proficiency (M = 5.06) was significantly lower than both L1 (t(143) = −44.16, p < .001) and L2 (t(143) = −42.07, p < .001).
Most notably, the writing dimension displayed a reverse dominance pattern, deviating from the historical usage hierarchy. L2 writing proficiency (M = 8.79) was significantly higher than L1 (M = 7.28), t(143) = 11.6, p < .001. Despite this reversal, both languages were still significantly more proficient than L3 (M = 5.60), as indicated by comparisons with L1 (t(143) = −12.9, p < .001) and L2 (t(143) = −124.4, p < .001).
Finally, we compared composite scores averaged across all four dimensions to assess overall ability. The analysis indicated that the specific domain variations balanced each other out: overall, there was no significant difference between the aggregate proficiency of L1 (M = 8.6075) and L2 (M = 8.695), t(143) = 1.33, p = .554. Consistent with the individual skills, overall L3 proficiency (M = 5.53) remained significantly lower than both L1 (t(143) = −46.66, p < .001) and L2 (t(143) = −47.99, p < .001).
We further explored subgroup differences based on individual dominance scores. This analysis confirmed that the vast majority of participants individually mirrored the aggregate pattern: displaying L1 dominance in speaking and reading, equivalent proficiency in listening, and L2 dominance in writing. Thus, while objective usage history favors L1, the functional proficiency data indicate that L2 capabilities have largely converged with L1 levels."
2. Revision regarding Section 3.2 (Page [14], Lines [542-550]): "3.2. Baseline Efficiencies of Production and Recognition Before examining the interactions, we assessed the independent characteristics of the production and recognition systems.
First, regarding Production Language, naming in an earlier-acquired language led to faster subsequent recognition overall (M = 893 ms) compared to a later-acquired language (M = 982 ms), t(131) = 6.00, p < .001, Cohen's d = 0.202.
Second, regarding Recognition Language, a significant effect was observed (ps < .001). Response times followed a proficiency-based pattern: L1 (888 ms) ≈ L2 (885 ms) < L3 (1040 ms)." |
||
|
Comments 5: [L369 What is the "cascade" is referred to here?] |
||
|
Response 5: We strictly appreciate the reviewer’s request for clarification regarding the term "cascade." We acknowledge that this term was somewhat metaphorical and lacked the necessary precision in this context, potentially leading to ambiguity.
In response to your comment, we have removed this vague expression from the revised manuscript. We have rephrased the relevant section to explicitly describe the underlying mechanism (i.e., the flow of activation or sequential processing) without relying on jargon. |
||
|
Comments 6: [L409 Please expand on how the authors, in their presentation of results, are able to see a "clear gradient contingent" ] |
||
|
Response 6: We appreciate the reviewer seeking clarification on this specific phrasing. Upon re-evaluating the text, we agree that the term "clear gradient contingent" was ambiguous and did not accurately reflect the statistical nuances of our findings. Consequently, we have removed this phrasing entirely from the revised manuscript. |
||
|
Comments 7: [Overall, the authors should reconsider most of the cases where they use "this" throughout the text, as what is referred to is not entirely clear. Please specify exactly what is referred to.] |
||
|
Response 7: We strictly appreciate the reviewer’s attention to the precision and clarity of our writing. We acknowledge that the use of the demonstrative pronoun "this" without an accompanying noun can create ambiguity regarding the intended referent.
In response to this suggestion, we have conducted a comprehensive review of the entire manuscript to identify and rectify these instances. We have systematically replaced vague occurrences of "this" with specific noun phrases to explicitly define the subject (e.g., revised to "this finding," "this interaction," "this discrepancy," or "this mechanism"). These revisions ensure that the logical flow and the specific elements being discussed are immediately clear to the reader. |
||
|
Comments 8: [ L360 Please refrain from the use of Foundational in the section title, if not previously defined as meaning something specific.] |
||
|
Response 8: We appreciate the reviewer's attention to the precision of our terminology. We agree that the term "Foundational" was not explicitly defined and could introduce unnecessary ambiguity regarding the nature of the evidence.
Accordingly, we have removed this term from the section title. The revised title now directly describes the content of the analysis (the main effect of cross-modal facilitation) without using undefined descriptors.
Changes in the manuscript (Page [14], Line [534]):
Revised Section Title: 3.1. Evidence for Cross-Modal Facilitation (Addressing RQ1) |
||
|
Comments 9: [L291 "is" should be "was".] |
||
|
Response 9: We thank the reviewer for this precise correction regarding tense consistency. We have amended the verb to the past tense ("was") to align with the descriptive style used throughout the Methods and Materials sections.
Changes in the manuscript:
Revised text: "A critical component of the experimental design was the creation of language-specific pseudowords that are pronounceable and orthographically plausible but semantically meaningless—to serve as novel foils in the experimental task." |
||
|
Comments 10: [ Please address formatting issues in Table 3.] |
||
|
Response 10: We apologize for the formatting inconsistencies in the previous version of the manuscript. We have completely reformatted Table 3 to ensure strict adherence to the journal’s guidelines and to enhance readability. Specifically, we have corrected the column alignments, adjusted the borders, and organized the fixed effects into clear subsections (Main Effects, Critical Interactions) for better visual clarity. |
||
Reviewer 3 Report
Comments and Suggestions for Authors
The paper One System, Two Rules: Asymmetrical Coupling of Speech Production and Reading Comprehension in the Trilingual Brain examines the interaction between production and comprehension in multilingual population, and it reaches conclusions about the differences in the top-down and bottom-up system in individuals who speak three languages.
Thank you for the opportunity to read and review this paper, which I consider an important contribution to the field.
I generally like the structure and the outline, as well as the overall idea and the goal of the study. However, some parts could be improved. Below I suggest some comments, and I hope the authors will find them relevant.
Introduction
This part is well written and it mentions relevant papers, models and gaps in the field. However, my impression is that the purpose is a bit vague or scattered. Is it to establish a baseline, is it to discuss the bottom-up / top-down mechanisms or is it really connected and primarily intended to serve as an aid in approaching multilingual patients with aphasia?
For example, the last sentence in the Introduction states: This mapping of asymmetrical activation pathways will provide the foundational model needed to interpret multilingual aphasia and to understand how the brain orchestrates the flow of information from speech production to reading comprehension.
Please elaborate more on the link between this and multilingual aphasia in the entire introduction. Highlight the purpose of the study more concretely. If this really is the purpose, then I think more is needed on the topic in the introduction. If this is just an additional contribution, then it should be stated as such. In general, more focus is needed to set the ground for the next sections.
Materials and Methods
Table 1: What exactly is Frequency of use (M), how is it measured, i.e. operationalized, is it minutes, hours? Please specify this.
Page 6, line 22: You state that 1 indicates no usage, and 5 indicates exclusive usage -> this information is missing in the table. Please be more specific and accurate in the presentation of the variables, methods and measures.
Materials:
More information on the materials used is needed in the paper. Which words were chosen, how were they controlled across languages? What were exactly the stimuli? This can significantly influence the results and the interpretations.
Results
The entire results section could be written in a more specific way, i.e. less scattered. I suggest making the Table 3 smaller and more readable, and organizing the sections / subtitles according to the research questions, so it is easier to follow.
Discussion
Discussion is overall written well, and the conclusions seem to be in place. However, again, I advise the authors to structure it in a way that it clearly follows the purpose - aims- questions, and finally the results. I suggest not making any comments or conclusions that fall out of the scope of the paper (which is maybe to be rewritten in the first place - see my comments about the Introduction). Make sure to know the focus and the exact questions, and discern the potential explanations from the additional contributions.
The same applies for the conclusion.
Again, thank you for a nice paper. I suggest to make the required changes if the authors agree with them. For this article to be published, the changes are needed, but I would like to highlight again the relevance of this paper in both research and clinical settings.
Sincerely ,
Reviewer
Author Response
For research article
|
Response to Reviewer 3 Comments
|
||||||||||||||||||||||||||||||||||||||||||||||||||||||||||||||||||||||||||||||||||||||
|
1. Summary |
|
|
||||||||||||||||||||||||||||||||||||||||||||||||||||||||||||||||||||||||||||||||||||
|
Thank you very much for taking the time to review this manuscript. Please find the detailed responses below and the corresponding revisions/corrections highlighted/in track changes in the re-submitted files.
|
||||||||||||||||||||||||||||||||||||||||||||||||||||||||||||||||||||||||||||||||||||||
|
2. Questions for General Evaluation |
Reviewer’s Evaluation |
Response and Revisions |
||||||||||||||||||||||||||||||||||||||||||||||||||||||||||||||||||||||||||||||||||||
|
Does the introduction provide sufficient background and include all relevant references? |
Can be improved |
|
||||||||||||||||||||||||||||||||||||||||||||||||||||||||||||||||||||||||||||||||||||
|
Is the research design appropriate? |
Yes |
|
||||||||||||||||||||||||||||||||||||||||||||||||||||||||||||||||||||||||||||||||||||
|
Are the methods adequately described? |
Can be improved |
|
||||||||||||||||||||||||||||||||||||||||||||||||||||||||||||||||||||||||||||||||||||
|
Are the results clearly presented? |
Yes |
|
||||||||||||||||||||||||||||||||||||||||||||||||||||||||||||||||||||||||||||||||||||
|
Are the conclusions supported by the results? |
Can be improved |
|
||||||||||||||||||||||||||||||||||||||||||||||||||||||||||||||||||||||||||||||||||||
|
Are all figures and tables clear and well-presented? |
Can be improved |
|
||||||||||||||||||||||||||||||||||||||||||||||||||||||||||||||||||||||||||||||||||||
|
3. Point-by-point response to Comments and Suggestions for Authors |
||||||||||||||||||||||||||||||||||||||||||||||||||||||||||||||||||||||||||||||||||||||
|
Comments 1: [This part is well written and it mentions relevant papers, models and gaps in the field. However, my impression is that the purpose is a bit vague or scattered. Is it to establish a baseline, is it to discuss the bottom-up / top-down mechanisms or is it really connected and primarily intended to serve as an aid in approaching multilingual patients with aphasia? For example, the last sentence in the Introduction states: This mapping of asymmetrical activation pathways will provide the foundational model needed to interpret multilingual aphasia and to understand how the brain orchestrates the flow of information from speech production to reading comprehension. Please elaborate more on the link between this and multilingual aphasia in the entire introduction. Highlight the purpose of the study more concretely. If this really is the purpose, then I think more is needed on the topic in the introduction. If this is just an additional contribution, then it should be stated as such. In general, more focus is needed to set the ground for the next sections. ]
|
||||||||||||||||||||||||||||||||||||||||||||||||||||||||||||||||||||||||||||||||||||||
|
Response 1: We sincerely thank the reviewer for this insightful observation. We agree that the initial draft may have attempted to cover too much ground, inadvertently obscuring the study's primary theoretical contribution.
To address this, we have extensively rewritten the Introduction to sharpen our focus. We have removed the peripheral discussion regarding clinical applications (e.g., aphasia) to prevent distraction. Instead, the revised text explicitly frames the study’s purpose as an investigation into the cognitive architecture of trilingualism, specifically aiming to disentangle the top-down mechanisms of speech production (governed by acquisition history) from the bottom-up mechanisms of reading comprehension (governed by social usage) (Lines 35–55).
We argue that trilingualism offers a unique methodological advantage to resolve the theoretical tension between developmental models (like the RHM) and usage-based models (like Multilink). The revised Introduction now builds a cohesive narrative leading directly to two competing hypotheses: the Age of Acquisition Primacy Hypothesis (focusing on the power of the prime) and the Social Usage Frequency Hypothesis (focusing on the receptivity of the target).
|
||||||||||||||||||||||||||||||||||||||||||||||||||||||||||||||||||||||||||||||||||||||
|
Comments 2: [Table 1: What exactly is Frequency of use (M), how is it measured, i.e. operationalized, is it minutes, hours? Please specify this. ]
|
||||||||||||||||||||||||||||||||||||||||||||||||||||||||||||||||||||||||||||||||||||||
|
Response 2: We thank the reviewer for requesting this clarification. We wish to specify that "Frequency of Use" was not measured in time-based units (e.g., minutes or hours) but was operationalized using a self-rated six-point Likert-type scale. Participants rated their language usage frequency in specific contexts on a scale from 0 to 5.
To address this, we have revised the relevant paragraph in the Participants section to explicitly define the scale anchors. Additionally, we have updated the Note in Table 1 to ensure the measurement unit is immediately clear to readers.
Changes in the manuscript:
1. Revised text in the Participants section (Lines 288–293): "With the functional equivalence between L1 and L2 confirmed by objective measures, we next sought to identify the sociolinguistic factors maintaining this equilibrium. To elucidate the environmental drivers of this bilingual balance, we examined current usage patterns across two distinct domains: home and social contexts. Usage frequency was operationalized using a six-point Likert-type scale rather than a duration-based measure. On this scale, a score of 0 indicated 'no usage' (Never), while a score of 5 indicated 'exclusive usage' (Always)."
2. Revised Note for Table 1: Note. Values are M (SD). Frequency of Use scores represent self-ratings on a six-point Likert-type scale (0–5), where 0 = never uses the language and 5 = uses the language exclusively/always. Home = within-family interactions; Social = interactions outside the home (friends, school/work, public). Residential Exposure = years living in regions where the language is the community language (English = 0 indicates no residence in English-speaking countries). Duration of Use = current age − age of acquisition, excluding self-reported non-use periods. MINT = Multilingual Naming Test, score range 0–68.
|
||||||||||||||||||||||||||||||||||||||||||||||||||||||||||||||||||||||||||||||||||||||
|
Comments 3: [Page 6, line 22: You state that 1 indicates no usage, and 5 indicates exclusive usage -> this information is missing in the table. Please be more specific and accurate in the presentation of the variables, methods and measures. ]
|
||||||||||||||||||||||||||||||||||||||||||||||||||||||||||||||||||||||||||||||||||||||
|
Response 3: We thank the reviewer for examining the presentation of our measures so closely. We apologize for the ambiguity regarding the rating scale anchors. To ensure precision and self-containment, we have revised the Note in Table 1 to explicitly state that "Frequency of Use" is operationalized on a 0–5 scale, where 0 indicates "never" and 5 indicates "exclusively/always." This update aligns the table directly with the methodology described in the text.
Changes in the manuscript:
Revised Table 1 (Content preserved, Note updated for clarity):
Table 1. Participant Language Background
Note. Values are M (SD). Frequency of Use scores are six-point Likert-type ratings (0–5; 0 = never … 5 = exclusively/always). Home = within-family interactions; Social = interactions outside the home (friends, school/work, public). Residential Exposure = years living in regions where the language is the community language (English = 0 indicates no residence in English-speaking countries). Duration of Use = current age − age of acquisition, excluding self-reported non-use periods. MINT = Multilingual Naming Test, score range 0–68. |
||||||||||||||||||||||||||||||||||||||||||||||||||||||||||||||||||||||||||||||||||||||
|
Comments 4: [More information on the materials used is needed in the paper. Which words were chosen, how were they controlled across languages? What were exactly the stimuli? This can significantly influence the results and the interpretations.]
|
||||||||||||||||||||||||||||||||||||||||||||||||||||||||||||||||||||||||||||||||||||||
|
Response 4: We appreciate the reviewer highlighting the need for greater transparency regarding our experimental materials. We agree that precise details regarding word selection, linguistic control, and stimulus construction are essential for the validity of the study. To address this, we have extensively revised Section 2.2.2 (Materials) (Lines 341-409). We now explicitly detail:
1.Word Selection and Control: We clarify that 48 concepts were selected from the International Picture Naming Project [27]. We have added data from a norming study (N=25) demonstrating that these items are balanced across English, Chinese, and Uyghur in terms of familiarity (p=.181) and picture-word matching accuracy (p=.677).
2.Stimulus Specification: We have provided a comprehensive description of the visual stimuli (pictures) and the construction of language-specific pseudowords (foils). We detail the use of Wuggy [53] and the CPN system [28] for generating foils and present statistical evidence confirming that real words and pseudowords were matched for orthographic complexity using standardized Z-scores.
3.Supplementary Data: We have added Appendix A (complete list of words and pseudowords) and Appendix B (sample picture stimuli) to provide full transparency.
|
||||||||||||||||||||||||||||||||||||||||||||||||||||||||||||||||||||||||||||||||||||||
|
Comments 5: [The entire results section could be written in a more specific way, i.e. less scattered. I suggest making the Table 3 smaller and more readable, and organizing the sections / subtitles according to the research questions, so it is easier to follow. ]
|
||||||||||||||||||||||||||||||||||||||||||||||||||||||||||||||||||||||||||||||||||||||
|
Response 5: We thank the reviewer for this constructive suggestion regarding the organization and presentation of our results. We agree that structuring the section by Research Question (RQ) significantly improves readability and logical flow. Accordingly, we have:
1.Restructured the Results Section: We have reorganized the text into three clear subsections. Section 3.1 addresses RQ1 (Cross-Modal Facilitation), Section 3.2 establishes the necessary baseline efficiencies, and Section 3.3 addresses RQ2 (Drivers of Asymmetry) (Line 534-561).
2.Simplified Table 3: We have condensed Table 3 to present only the critical fixed effects and interactions relevant to the main discussion. The complete model parameters, including all pairwise comparisons and non-significant higher-order interactions, have been moved to Appendix C (Table C1) to declutter the main text.
|
||||||||||||||||||||||||||||||||||||||||||||||||||||||||||||||||||||||||||||||||||||||
|
Comments 6: [Discussion is overall written well, and the conclusions seem to be in place. However, again, I advise the authors to structure it in a way that it clearly follows the purpose - aims- questions, and finally the results. I suggest not making any comments or conclusions that fall out of the scope of the paper (which is maybe to be rewritten in the first place - see my comments about the Introduction). Make sure to know the focus and the exact questions, and discern the potential explanations from the additional contributions. The same applies for the conclusion. ] |
||||||||||||||||||||||||||||||||||||||||||||||||||||||||||||||||||||||||||||||||||||||
|
Response 6: We appreciate the reviewer’s positive assessment of the Discussion and the constructive advice regarding its structure. We agree that a stricter alignment with our initial Research Questions (RQs) enhances the clarity of the manuscript. Therefore, we have extensively revised Section 4 (Discussion) (Line 562-659) and Section 5 (Conclusions) (Line 670-679).
Specifically, we have: 1.Restructured the Discussion: The sections are now explicitly organized to mirror the research questions. Section 4.1 discusses the general architecture (RQ1), and Section 4.2 addresses the specific drivers of asymmetry (RQ2).
2.Refined the Scope: Following the reviewer’s suggestion to avoid out-of-scope comments, we have removed the speculative discussion regarding aphasia. Instead, in Section 4.3, we focus strictly on the methodological contributions inherent to our specific trilingual design.
3.Clarified the Conclusion: Section 5 has been tightened to ensure it summarizes the findings solely within the context of the "Dual Mechanism" framework established by our results.
|
||||||||||||||||||||||||||||||||||||||||||||||||||||||||||||||||||||||||||||||||||||||

Reviewer 4 Report
Comments and Suggestions for Authors
This manuscript investigates cross-linguistic priming between speech production and reading comprehension in Uyghur–Chinese–English trilinguals. Using a production-to-comprehension paradigm, the authors test how age of acquisition (AoA), proficiency, and usage patterns modulate cross-linguistic activation. The study is ambitious, theoretically motivated, and addresses a genuine gap in multilingualism research, especially regarding baseline models relevant to multilingual aphasia. The methodological framework is sound, and the paper is rich in theoretical integration.
However, the manuscript is at times difficult to follow due to conceptual overextension, repetition, and lack of clarity in argument structure. Several methodological and interpretive issues require clarification. Statistical reporting is not entirely consistent, and crucial details on design, counterbalancing, pseudoword matching, and exclusion criteria need elaboration. The Discussion, while insightful, is too expansive and speculative in parts. Substantial streamlining is recommended. Additionally, although the manuscript frequently references aphasia as a potential application, the study does not directly investigate aphasia and does not include clinical populations. Any claims about aphasia are therefore speculative and should be framed more cautiously.”
Overall, the work is promising but requires significant revision before publication.
Major comments
- Introduction
Line 41–51: Consider simplifying multilingual control discussion; some sentences are repetitive.
Line 56–71: Aphasia recovery patterns are explained in too much detail for an introduction not focusing on clinical populations.
Lines 35–153; 423–571
In the Introduction and Discussion are not related to the aim of the study. The Discussion in particular presents theoretical arguments (embodiment, aphasia recovery patterns, inhibitory control, sensory-motor integration) that extend far beyond what the experimental results warrant.
It is suggested to focus on claims directly supported by the data.
- Clarify the rationale and predictions
Lines 134–153
The AoA Primacy vs. Language Dominance hypotheses are well motivated, but predictions remain vague. For example, does AoA predict monotonic decrease L1 > L2 > L3 for all prime–target pairings, or only when L2 or L3 are targets? Similarly, does dominance predict symmetric or asymmetric priming when L2 is the prime?
- Methods
Lines 270–275
The construction of pseudowords in Chinese vs. alphasyllabaries (Uyghur) vs. alphabetic (English) differs dramatically. This asymmetry may influence lexical decision difficulty. Provide evidence that pseudowords were matched on difficulty or discuss this limitation.
Lines 267–289
It is not clear how picture sets A/B were counterbalanced across all six language-pair conditions. Also, “two experimental series” is vague—were these fully counterbalanced across participants?
- Statistics
Lines 316–334; 348–350
- No participant random effects are included. Even in between-subjects designs, subject intercepts are often advisable due to within-subject RT variance.
- The model includes only item intercepts, but no slopes for lexical status by item (usually necessary for priming).
- Reaction time trimming: ±3 SD is acceptable, but justification is minimal.
- Effect sizes (Cohen’s d) reported for fixed effects in LMMs need clarification—how were they computed?
- The paper reports t-values but does not provide degrees of freedom.
Explain why participant random effects were omitted, and provide more detail on the effect-size computation. Consider including maximal or near-maximal random structure unless convergence issues prevented it (and report those issues).
- Discussion
Lines 423–520
The Discussion attributes patterns to embodied cognition, conceptual grounding, domain specificity of home vs. social usage, and task-dependent inhibition. These are theoretically interesting, but the behavioral data alone do not strongly support these claims.
Lines 52–71; 437–439; 537–547
The manuscript recurrently claims that the results establish a baseline for multilingual aphasia interpretation. But no clinical data are included, and the connection remains largely hypothetical. Reframe this claim as a potential contribution rather than a direct outcome.
The manuscript addresses an important gap in multilingual psycholinguistics and presents novel empirical findings. However, substantial revisions are necessary for clarity, methodological transparency, and theoretical precision. With significant restructuring and focused argumentation, the study has strong potential for publication.
Author Response
For research article
|
Response to Reviewer 4 Comments
|
||
|
1. Summary |
|
|
|
Thank you very much for taking the time to review this manuscript. Please find the detailed responses below and the corresponding revisions/corrections highlighted/in track changes in the re-submitted files.
|
||
|
2. Questions for General Evaluation |
Reviewer's Evaluation |
Response and Revisions |
|
Does the introduction provide sufficient background and include all relevant references? |
Must be improved
|
|
|
Is the research design appropriate? |
Can be improved |
|
|
Are the methods adequately described? |
Can be improved |
|
|
Are the results clearly presented? |
Can be improved |
|
|
Are the conclusions supported by the results? |
Must be improved
|
|
|
Are all figures and tables clear and well-presented? |
Can be improved |
|
|
3. Point-by-point response to Comments and Suggestions for Authors |
||
|
Comments 1: [Line 41–51: Consider simplifying multilingual control discussion; some sentences are repetitive.]
|
||
|
Response 1: We thank the reviewer for this observation. We agree that the original discussion regarding general control mechanisms was somewhat repetitive and overly broad. We have significantly streamlined this section to focus directly on the core theoretical puzzle: the asymmetry of cross-linguistic influence. The revised text now moves swiftly from the general premise of language non-selectivity to the specific problem of disentangling Age of Acquisition (AoA) from Social Usage Frequency, highlighting the necessity of the trilingual design (Line 35-166).
|
||
|
Comments 2: [Line 56–71: Aphasia recovery patterns are explained in too much detail for an introduction not focusing on clinical populations.]
|
||
|
Response 2: We appreciate the reviewer's feedback regarding the relevance of the clinical literature. We agree that the detailed discussion of aphasia recovery patterns was beyond the scope of this behavioral study on healthy trilingual speakers and interrupted the flow of the theoretical argument. Consequently, we have deleted this section entirely, along with the associated references. The Introduction now transitions directly from the theoretical debate regarding lexical activation to the methodological limitations of bilingual models, ensuring a tighter focus on the cognitive mechanisms of the multilingual lexicon.
|
||
|
Comments 3: [Lines 35–153; 423–571 In the Introduction and Discussion are not related to the aim of the study. The Discussion in particular presents theoretical arguments (embodiment, aphasia recovery patterns, inhibitory control, sensory-motor integration) that extend far beyond what the experimental results warrant.
It is suggested to focus on claims directly supported by the data.]
|
||
|
Response 3: We sincerely thank the Reviewer for this critical and insightful observation. We fully agree that the initial inclusion of theoretical frameworks regarding aphasia recovery, embodied cognition, and inhibitory control extended beyond the scope of our experimental findings. Since our data revealed robust cross-linguistic facilitation (priming) rather than interference, discussing inhibitory mechanisms was indeed speculative and not directly warranted by the results.
Therefore, we have substantially revised both the Introduction and Discussion sections to align strictly with the claims supported by our data. Specifically:
Removal of Extraneous Theories: We have completely removed the sections discussing aphasia, sensory-motor integration, and embodied cognition.
Exclusion of Inhibitory Control: We have removed the discussion on inhibitory control. As our results demonstrate a global facilitatory effect, utilizing an inhibitory framework was unnecessary and potentially confusing.
Refocused Narrative: The manuscript now focuses exclusively on the two factors that were experimentally manipulated and statistically significant in our results: Age of Acquisition (AoA) and Social Usage Frequency.
|
||
|
Comments 4: [Lines 134–153 The AoA Primacy vs. Language Dominance hypotheses are well motivated, but predictions remain vague. For example, does AoA predict monotonic decrease L1 > L2 > L3 for all prime–target pairings, or only when L2 or L3 are targets? Similarly, does dominance predict symmetric or asymmetric priming when L2 is the prime?]
|
||
|
Response 4: We appreciate the Reviewer's request for greater precision regarding our theoretical predictions. We agree that the initial formulation did not sufficiently distinguish how each hypothesis specifically modulates the prime versus the target.
To address this, we have rewritten the Introduction to explicitly operationalize the predictions of the Age of Acquisition (AoA) Primacy Hypothesis and the Social Usage Frequency Hypothesis. We now clarify that:
The AoA Hypothesis predicts that the prime's efficacy is a function of acquisition order (predicting L1 prime > L3 prime when the target is held constant) (line 132-145).
The Social Usage Hypothesis predicts that the target's sensitivity is a function of social usage frequency (predicting Target L2 > Target L3 when the prime is held constant) (line 146-159).
|
||
|
Comments 5: [Lines 270–275 The construction of pseudowords in Chinese vs. alphasyllabaries (Uyghur) vs. alphabetic (English) differs dramatically. This asymmetry may influence lexical decision difficulty. Provide evidence that pseudowords were matched on difficulty or discuss this limitation.]
|
||
|
Response 5: We thank the Reviewer for raising this crucial point regarding the inherent asymmetry in constructing pseudowords across logographic (Chinese) and alphabetic/alphasyllabic (English/Uyghur) scripts. We fully acknowledge that these script-specific differences could potentially confound lexical decision difficulty. To address this, we implemented a rigorous validation process involving both behavioral pre-testing and statistical matching of orthographic complexity. We have also added a discussion on the limitations regarding visual processing asymmetries.
1. Stimulus Validation and Behavioral Evidence To ensure the pseudowords functioned correctly as "plausible but non-existent" lexical items, all materials were first validated by 25 Uyghur-Chinese-English trilinguals who did not participate in the main experiment. In the main study, the low error rates across all languages (≤ 4.34%) confirmed that participants could reliably distinguish these validated pseudowords from real words, suggesting that difficulty levels were manageable and comparable.
2. Statistical Control of Orthographic Complexity We conducted a series of analyses to verify that real words and pseudowords were matched on physical complexity within and across languages.
Raw Orthographic Units: A 2 (Lexicality: Real vs. Pseudo) × 3 (Language) ANOVA on the number of basic units (letters for Uyghur/English; strokes for Chinese) revealed the expected main effect of Language (F(2,282)=102.776,p<.001,η2=0.422), reflecting the higher stroke count of Chinese characters. Crucially, however, there was no significant main effect of Lexicality (F(1,282)=0.229,p=.632) and no Lexicality × Language interaction (F(2,282)=0.108,p=.898). This confirms that within each language, pseudowords were structurally equivalent to real words.
Standardized Complexity: To allow for a direct cross-linguistic comparison, we converted length values to z-scores within each language. A subsequent ANOVA on these standardized scores eliminated the Language effect (F(2,282)=0.0001,p=.998). Importantly, the main effect of Lexicality (F(1,282)=0.24,p=.625) and the interaction (F(2,282)=0.0255,p=.975) remained non-significant, providing robust evidence that stimulus length was equivalent across conditions relative to script norms.
Syllabic Structure: We further verified matching at the syllable level. A 2 × 3 ANOVA on syllable counts showed no significant main effect for Lexicality (F(1,282)=0.229,p=.632) and no interaction (F(2,282)=0.00828,p=.928).
3. Analytical Strategy and Limitations Analytically, we mitigated baseline differences by focusing on priming effects (reaction time differences) rather than absolute reaction times. This relative measure serves as an internal control for the varying processing speeds inherent to different scripts.
However, we agree that visual processing asymmetries cannot be entirely eliminated. We have added the following text to the Discussion/Limitations section to explicitly acknowledge this:
"Finally, a key challenge is the inherent visual processing asymmetry across scripts (e.g., logographic Chinese vs. alphabetic Uyghur/English). While we mitigated this by using relative priming effects as our main metric, this constraint remains. Future studies could more precisely calibrate cross-linguistic comparisons by establishing script-specific processing baselines, thereby enhancing the validity of the findings."
|
||
|
Comments 6: [Lines 267–289 It is not clear how picture sets A/B were counterbalanced across all six language-pair conditions. Also, “two experimental series” is vague—were these fully counterbalanced across participants?]
|
||
|
Response 6: We thank the Reviewer for pointing out the ambiguity regarding our counterbalancing procedures. We apologize for the confusion caused by the term "experimental series." We have revised the Method section to explicitly describe how the picture sets (A and B) were rotated to create two fully counterbalanced lists (List 1 and List 2). We clarify that this rotation occurred within each of the six language-pair conditions, ensuring that every target item served equally as a "studied" (primed) and "unstudied" (baseline) stimulus across the participant pool.
Revised Manuscript Text:
To operationalize the crucial within-subjects factor of Lexical Status, the experimental procedure was divided into two distinct phases: a production phase (picture naming) followed by a recognition phase (lexical decision task). The stimuli for these tasks consisted of 48 experimental pictures sourced from the International Picture Naming Project database [27]. To control for item-specific confounds, these pictures were first partitioned into two equivalent sets (Set A and Set B, n = 24 each), which were then used to create two master experimental lists (List 1 and List 2). Within each of the six language conditions, participants were randomly assigned to one of these lists (n = 12 per list), a step that determined which items would serve as 'studied' versus 'unstudied'. For instance, in the Chinese→English condition, a participant assigned to List 1 would name a picture of an apple in Chinese ("苹果"), making the English word "apple" a 'studied' item. For a participant in List 2, "apple" would be an 'unstudied' item. This fully counterbalanced design ensures every item appeared equally often in both conditions, isolating any calculated priming effect from item-specific properties. To construct the final recognition phase, these 48 target words (24 studied, 24 unstudied) were presented alongside an equal number of 48 pronounceable, language-specific pseudowords (e.g., driss in English) that served as foils. The presentation of target words and pseudowords was randomly intermixed for each participant, as was the order of all trials within each phase.
|
||
|
Comments 7: [No participant random effects are included. Even in between-subjects designs, subject intercepts are often advisable due to within-subject RT variance. The model includes only item intercepts, but no slopes for lexical status by item (usually necessary for priming).]
|
||
|
Response 7: We sincerely thank the Reviewer for this critical statistical insight. We fully agree that the initial model specification was insufficient for capturing the variance inherent in reaction time data. To address this, we have re-analyzed the dataset using a maximal random effects structure, following the best-practice recommendations of Barr et al. [56].
The revised model now explicitly accounts for the non-independence of observations and ensures generalizability across both populations of participants and items. Specifically:
1.Random Intercepts: We included random intercepts for both participants (SubjectID) and items (ItemID) to control for baseline differences in response speed.
2.Random Slopes: Crucially, addressing the Reviewer's specific concern regarding priming mechanisms, we included by-participant and by-item random slopes for Lexical Status. This allows the model to account for individual variation in the magnitude of the priming effect and item-specific sensitivities.
The final model formula was specified as follows: RT ~ Lexical_Status * Recognition_Language * Production_Language + (1 + Lexical_Status | SubjectID) + (1 + Lexical_Status | ItemID)
Revised Manuscript Text (Method/Results Section):
"To account for the non-independence of observations and to ensure the results generalize to the broader populations of participants and items, we adopted a maximal random effects structure as recommended by Barr et al. [56]. The final model included random intercepts for both participants (SubjectID) and items (ItemID). Crucially, to control for individual variation in the magnitude of the priming effect, we also included by-participant and by-item random slopes for the within-unit factor, Lexical Status. The final model formula was specified as: RT ~ Lexical_Status * Recognition_Language * Production_Language + (1 + Lexical_Status | SubjectID) + (1 + Lexical_Status | ItemID)."
|
||
|
Comments 8: [Reaction time trimming: ±3 SD is acceptable, but justification is minimal.]
|
||
|
Response 8: We appreciate the Reviewer's feedback regarding the data cleaning procedure. We have revised the text to more explicitly articulate the methodological and statistical grounds for employing the ±3 SD cutoff.
While we adhere to standard practices in reaction time research, our justification is two-fold:
Cognitive Validity: Following Ratcliff [60], extreme outliers typically reflect processes distinct from the cognitive mechanism of interest (e.g., momentary attentional lapses or premature motor responses) rather than genuine slow or fast processing.
Statistical Assumptions: As noted by Baayen and Milin [61], Linear Mixed-Effects Models (LMMs) operate on the assumption that model residuals are normally distributed. Trimming extreme values is a necessary step to satisfy this requirement and prevent the model from being skewed by non-representative data points.
Revised Manuscript Text:
"Prior to analysis, trials with reaction times deviating by more than ±3 standard deviations from each participant's mean were excluded. This specific criterion was applied to isolate genuine cognitive processing from outliers attributable to non-cognitive factors, such as attentional lapses or anticipatory motor errors [60]. Furthermore, this trimming procedure was essential to ensure that the data conformed to the normality assumptions governing residual distributions required for valid linear mixed-effects modeling [61]. Following this data cleaning process, 4.34% of the total observations were removed."
|
||
|
Comments 9: [Effect sizes (Cohen's d) reported for fixed effects in LMMs need clarification—how were they computed?]
|
||
|
Response 9: We appreciate the Reviewer's request for clarification regarding the calculation of effect sizes, as we recognize that methods for deriving Cohen's d from Linear Mixed Models can vary.
In our analysis, we computed Cohen's d to quantify the standardized mean difference for the fixed effects and planned contrasts. We have revised the methodology section to explicitly state that d was calculated by dividing the difference between the estimated marginal means (EMMs) by the square root of the model's residual variance (i.e., the residual standard deviation, σ). This approach focuses on the effect relative to the unexplained variance at the observation level, following the method described by Westfall et al. [63] and implemented via the emmeans package.
Revised Manuscript Text:
"Statistical significance for fixed effects was evaluated using Satterthwaite's method [57] for approximating degrees of freedom, as implemented in the lmerTest package [58]. This approach was chosen for its established reliability in maintaining Type I error rates within acceptable limits for psycholinguistic data [59].
To interpret the magnitude of the observed effects, we calculated Cohen's d for all fixed effects and pairwise comparisons using the emmeans package [62, 64]. Specifically, the effect size was derived by standardizing the difference between the estimated marginal means (EMMs) against the square root of the model's residual variance (σ), a method recommended for characterizing effect magnitudes in mixed-effects models [63]."
|
||
|
Comments 10: [ The paper reports t-values but does not provide degrees of freedom.]
|
||
|
Response 10: We thank the Reviewer for noting this omission. We agree that reporting degrees of freedom is essential for the complete interpretation of the t-statistics derived from Linear Mixed Models. We have thoroughly revised the Results section to include the approximate degrees of freedom for all t-tests (estimated using Satterthwaite's method).
Below is an excerpt from the revised results demonstrating this update:
Revised Manuscript Text:
"The analysis revealed a significant and robust cross-language priming effect: responses to studied words (M = 863 ms) were significantly faster than to unstudied words (M = 1012 ms), t(126) = 35.5, p < .001, Cohen's d = 0.431.
Regarding Production Language, naming in an earlier-acquired language led to faster subsequent recognition overall (M = 893 ms) compared to a later-acquired language (M = 982 ms), t(131) = 6.00, p < .001, Cohen's d = 0.202.
Furthermore, a significant interaction between Language and Lexical Status was observed, t(126) = 5.72, p < .001, Cohen's d = 0.401."
|
||
|
Comments 11: [Lines 423–520 The Discussion attributes patterns to embodied cognition, conceptual grounding, domain specificity of home vs. social usage, and task-dependent inhibition. These are theoretically interesting, but the behavioral data alone do not strongly support these claims.]
|
||
|
Response 11: We sincerely thank the Reviewer for this insightful observation. We agree that the initial interpretation extended beyond the scope of the behavioral data, particularly regarding theoretical constructs such as embodied cognition and conceptual grounding.
In response, we have substantially revised the Discussion section (Section 4). We have removed the speculative passages concerning embodied cognition, conceptual grounding, and task-dependent inhibition. The revised discussion now strictly adheres to the empirical findings, focusing on the dissociation between Age of Acquisition (AoA) as a driver of production-based priming and Social Usage Frequency as a determinant of recognition efficiency. We believe this focused interpretation provides a more robust and data-driven account of the observed asymmetries in the trilingual lexicon.
Revised Manuscript Text:
4. Discussion
4.1. The Integrated and Asymmetrical Architecture of the Trilingual Lexicon (Addressing RQ1) The primary objective of this study was to elucidate the functional architecture linking speech production and reading comprehension within the trilingual brain. Specifically, we investigated whether the motor act of naming facilitates the subsequent recognition of translation equivalents. Our results provide a clear affirmative answer: significant cross-language priming was observed across all languages. This confirms that the trilingual lexicon operates as an integrated system characterized by global co-activation [29, 30].
However, this co-activation is not uniform. Our central finding reveals a fundamental functional dissociation. The top-down influence (production-to-comprehension) is primarily driven by Age of Acquisition (AoA), reflecting the strength of concept-to-lexicon links established early in life. Conversely, the bottom-up efficiency of the recognition system is governed by social usage frequency, reflecting the dynamic resting-state activation of lexical representations. This dissociation suggests that cross-linguistic interactions are asymmetrically weighted rather than symmetrical [31, 32]. Ultimately, these findings indicate that the trilingual lexicon is not a monolithic entity but a complex system that balances stable historical constraints (AoA) with dynamic environmental demands (social usage).
4.2. Resolving the Theoretical Tension: AoA Primacy vs. Social Usage (Addressing RQ2) Our second research question (RQ2) sought to resolve the tension between the AoA Primacy Hypothesis (the power of the prime) and the Social Usage Frequency Hypothesis (the receptivity of the target). Our data suggest these are not mutually exclusive alternatives but distinct mechanisms governing different ends of the processing chain.
4.2.1. The Power of the Prime: Support for AoA Primacy Regarding the top-down influence originating from speech production, our data identify AoA as the primary determinant. Naming in an earlier-acquired language (L1 or L2) generated significantly larger cross-language priming effects than using the later-acquired L3. This aligns with developmental frameworks like the Revised Hierarchical Model (RHM), which posit that early language learning establishes more robust conceptual-lexical links [10, 11]. Since picture naming requires retrieving a lemma directly from a non-linguistic concept, early-acquired languages appear to trigger a more improved spreading of activation. This is consistent with neurocognitive research indicating that L1 processing elicits deeper semantic engagement than formal L2/L3 instruction [35]. Thus, the "power" of the prime is a function of acquisition history: early-forged links create a resonant signal that propagates effectively to other languages. Importantly, this facilitation was global—using a dominant language facilitated, rather than inhibited, the recognition of other languages.
4.2.2. The Receptivity of the Target: Support for Social Usage Frequency In contrast, the bottom-up recognition system followed principles governed by social usage patterns. While baseline reaction times reflected standard proficiency effects (L1/L2 > L3) [39, 40], the magnitude of the priming benefit revealed a distinct hierarchy: L2 Chinese (socially dominant) showed the largest priming effect, followed by L1 Uyghur, with L3 English showing the least. This L2 > L1 > L3 pattern challenges the assumption that "frequency of use" is a monolithic variable.
Crucially, Home Usage (highest for L1) did not predict the priming hierarchy, whereas Social Usage (highest for L2) mirrored the observed magnitude. This distinction draws upon the functional specialization of usage domains [24]. We propose that Social Usage acts as the primary driver for tuning resting-level activation thresholds. Unlike the often predictable context of home usage, social environments involve diverse interactions that may more effectively raise activation thresholds [41–43]. Consequently, L2 words appear to act as more sensitive "receivers." When the signal arrives from the prime, the highly activated L2 representations reach recognition thresholds faster. While consistent with the global co-activation posited by the BLINCS model [44], these results highlight the need for theoretical frameworks that move beyond binary bilingual models to capture the specific complexities of the trilingual experience [45–49].
4.3. Methodological Contributions: The Utility of the Trilingual Design This study underscores the unique methodological value of trilingualism in psycholinguistics. In traditional bilingual research, disentangling AoA from Usage is difficult because switching translation direction (e.g., L1→L2 vs. L2→L1) simultaneously alters both the source and target languages.
Our "triangular" design overcomes this limitation. By holding the production language (AoA of the prime) constant while varying the comprehension language (Social Usage of the target)—for instance, comparing L1→L2 against L1→L3—we effectively disentangled the contribution of the speech motor system from that of the reading system. This approach demonstrates that information flow is context-sensitive, dynamically recruiting historical conceptual links or current lexical activation to meet the distinct demands of speaking and reading. This validates the argument that trilingualism offers a distinct experimental state, providing a clearer window into the component processes of language control than binary models alone.
|
||
|
Comments 11: [Lines 52–71; 437–439; 537–547 The manuscript recurrently claims that the results establish a baseline for multilingual aphasia interpretation. But no clinical data are included, and the connection remains largely hypothetical. Reframe this claim as a potential contribution rather than a direct outcome.]
|
||
|
Response 11: We appreciate the Reviewer’s valid point regarding the extrapolation of our findings to clinical populations. We agree that, in the absence of clinical data, drawing direct connections to multilingual aphasia remains speculative and extends beyond the scope of our behavioral results. Consequently, to ensure the manuscript remains strictly grounded in the empirical evidence, we have removed all passages discussing multilingual aphasia (specifically Lines 52–71, 437–439, and 537–547). The revised manuscript now focuses exclusively on the cognitive mechanisms of cross-linguistic priming in healthy trilingual speakers.
|
||
Round 2
Reviewer 4 Report
Comments and Suggestions for Authors
I thank the authors for their thorough and thoughtful revision. The manuscript has improved substantially in focus, methodological transparency, and alignment between theory and data.
They have implemented all major revisions. They removed the speculative material on aphasia, embodied cognition, and inhibitory control, and refocused both the Introduction and Discussion strictly on the AoA–vs.–Social Usage framework. They rewrote the theoretical section to provide clear, testable predictions for prime-driven AoA effects and target-driven usage effects. The stimuli section now includes full pseudoword validation and orthographic-complexity checks, with an explicit acknowledgment of remaining cross-script limitations. The experimental design has been clarified through a transparent description of picture-set counterbalancing. The statistical analysis was strengthened via a maximal LMM structure with appropriate random effects, and the authors clarified the RT-trimming rationale, effect-size computation, and degrees of freedom reporting. Finally, all unsupported clinical claims about multilingual aphasia have been removed.
I have only a few minor suggestions that the authors may wish to consider.
Minor suggestions
- Some effects described as “robust” have relatively small effect sizes (e.g., Cohen’s d around 0.2). While these are statistically reliable, the authors might slightly soften the language (e.g., “reliable” or “modest but consistent”) to better match the reported magnitudes.
- There are a few long sentences in the Introduction and Discussion that could be broken into two for readability, but this is purely stylistic and does not affect scientific content.
Author Response
|
Response to Reviewer 4 Comments
|
||
|
1. Summary |
|
|
|
Thank you very much for taking the time to review this manuscript. Please find the detailed responses below and the corresponding revisions/corrections highlighted/in track changes in the re-submitted files.
|
||
|
2. Questions for General Evaluation |
Reviewer's Evaluation |
Response and Revisions |
|
Does the introduction provide sufficient background and include all relevant references? |
Yes |
|
|
Is the research design appropriate? |
Yes |
|
|
Are the methods adequately described? |
Yes |
|
|
Are the results clearly presented? |
Yes |
|
|
Are the conclusions supported by the results? |
Yes |
|
|
Are all figures and tables clear and well-presented? |
Yes |
|
|
3. Point-by-point response to Comments and Suggestions for Authors |
||
|
Comments 1: [Some effects described as "robust" have relatively small effect sizes (e.g., Cohen's d around 0.2). While these are statistically reliable, the authors might slightly soften the language (e.g., "reliable" or "modest but consistent") to better match the reported magnitudes.]
|
||
|
Response 1: We appreciate this precise and helpful observation. We agree that while our findings are statistically significant, the effect sizes (particularly for the main effect of Production Language, where Cohen's d = 0.202) warrant more measured descriptors.
Accordingly, we have carefully reviewed the manuscript and calibrated the language to better reflect the magnitude of the observed effects. We have replaced terms like "robust" with "reliable," "effective," or "consistent" to ensure a scientifically rigorous presentation of the data.
Specific revisions include:
1.Abstract: We modified the description of the priming signals to emphasize effectiveness rather than strength.
Revised text: "...earlier-acquired languages (specifically L1) generated more effective priming signals than L2."
2.Results (Page 14, lines 538–539): We replaced "robust" with "reliable" when reporting the main priming effect.
Revised text: "The analysis revealed a significant and reliable cross-language priming effect: responses to studied words (M = 863 ms) were significantly faster than to unstudied words (M = 1012 ms), t(126) = 35.5, p < .001, Cohen's d = 0.431."
3.Discussion (Page 15, lines 599–601): We softened the language regarding the influence of early-acquired languages.
Revised text: "The finding that early-acquired languages produce a consistent facilitation suggests that they trigger a more effective spreading of activation from the conceptual level."
4.Discussion (Page 15, lines 604–605): We adjusted the terminology regarding the top-down signal.
Revised text: "...early-forged links create a reliable signal that propagates effectively to other languages."
|
||
|
Comments 2: [There are a few long sentences in the Introduction and Discussion that could be broken into two for readability, but this is purely stylistic and does not affect scientific content.]
|
||
|
Response 2: We thank the reviewer for this helpful suggestion regarding the stylistic presentation. We agree that readability is crucial for effectively communicating our findings. We have carefully reviewed the manuscript, with a specific focus on the Introduction and Discussion sections, and have broken down several complex sentences into shorter, clearer statements to improve the flow.
Specific examples of these revisions include:
1. Introduction (Page 2, lines 47–51): We split the sentence regarding the limitations of bilingual studies to clarify the "methodological bottleneck."
Revised text: "While bilingual studies have provided the foundation for these questions, they often face a methodological bottleneck. In bilinguals, the 'dominant' language is usually both the native language (L1) and the most frequently used one. This confound makes it difficult to disentangle the effects of Age of Acquisition (AoA) from Social Usage Frequency."
2. Introduction (Page 2, lines 52–56): We separated the study's methodology from its specific aim.
Revised text: "The present study leverages a trilingual population to dissociate these factors. Specifically, we aim to determine whether the flow of activation from speech production to reading comprehension is governed by the static stability of early acquisition [10] or the dynamic accessibility of current social use [12, 13]."
3. Introduction (Page 2, lines 65–68): We divided the sentence discussing usage-based models to better emphasize the supporting evidence.
Revised text: "On the other side, usage-based models like Multilink argue that current language dominance and frequency of use are more potent modulators [12]. This view is supported by findings that L2 use is a better predictor of L2→L1 priming than proficiency alone [13]."
4. Introduction (Page 3, lines 90–96): We broke down the complex description of L3 acquisition models.
Revised text: "This question is impossible to address within a bilingual paradigm. In contrast, trilingualism allows us to investigate critical new questions, such as whether the influence of an L3 on an L2 operates differently than that of an L1. The complexity of L3 acquisition is evident in the diverse models proposed to explain its trajectory. These models implicate a wide range of factors, from the persistent influence of the L1 to the privileged role of the L2 and the variable of typological proximity (e.g., [17, 18])."
5. Discussion (Page 15, lines 574–577): We simplified the statement regarding the functional dissociation found in our results.
Revised text: "Our central finding reveals a fundamental functional dissociation. The top-down influence flowing from production-to-comprehension is consistently driven by Age of Acquisition (AoA). This pattern reflects the unique strength of concept-to-lexicon links forged in early life."
6. Discussion (Page 16, lines 625–628): We split the sentence explaining the role of Social Usage to clarify the interpretation.
Revised text: "We demonstrate that Social Usage is the primary driver. This suggests that the contextual diversity inherent in social interaction is more effective at raising activation thresholds than the predictable context of home usage [41–43]."
7. Conclusions (Page 17, lines 676–681): We refined the concluding remarks to distinctively separate the production and recognition mechanisms.
Revised text: "On the one hand, production is history-driven: it operates as a top-down process governed by the AoA Primacy principle. Here, the privileged conceptual access of the earliest-acquired language generates a distinct signal. On the other hand, recognition is environment-driven: it functions as a bottom-up process governed by the Social Usage Frequency principle. In this system, the demands of social interaction determine the resting-level receptivity of the lexicon."
|
||